# Reliable Estimation of KL Divergence using a Discriminator in Reproducing Kernel Hilbert Space

**Sandesh Ghimire , Aria Masoomi, Jennifer Dy**
Department of Electrical and Computer Engineering
Northeastern University
sandesh@ece.neu.edu, masoomi.a@northeastern.edu, jdy@ece.neu.edu

## Abstract

Estimating Kullback–Leibler (KL) divergence from samples of two distributions is essential in many machine learning problems. Variational methods using neural network discriminator have been proposed to achieve this task in a scalable manner. However, we noted that most of these methods using neural network discriminators suffer from high fluctuations (variance) in estimates and instability in training. In this paper, we look at this issue from statistical learning theory and function space complexity perspective to understand why this happens and how to solve it. We argue that the cause of these pathologies is lack of control over the complexity of the neural network discriminator function space and could be mitigated by controlling it. To achieve this objective, we 1) present a novel construction of the discriminator in the Reproducing Kernel Hilbert Space (RKHS), 2) theoretically relate the error probability bound of the KL estimates to the complexity of the discriminator in the RKHS space, 3) present a scalable way to control the complexity (RKHS norm) of the discriminator for a reliable estimation of KL divergence, and 4) prove the consistency of the proposed estimator. In three different applications of KL divergence – estimation of KL, estimation of mutual information and Variational Bayes – we show that by controlling the complexity as developed in the theory, we are able to reduce the variance of KL estimates and stabilize the training.

## 1 Introduction

Estimating Kullback–Leibler (KL) divergence from data samples is an essential component in many machine learning problems including Bayesian inference, calculation of mutual information or methods using information theoretic objectives. Variational formulation of Bayesian Inference requires KL divergence computation, which could be challenging when we only have finite samples from two distributions. Similarly, computation of information theoretic objectives like mutual information requires computation of KL divergence between the joint and the product of marginals.

KL divergence estimation from samples was studied thoroughly by Nguyen et al. [1] using a variational technique, convex optimization and RKHS norm regularization, while also providing theoretical guarantees and insights. However, their technique requires handling the whole dataset at once and is not scalable. Many modern models need to use KL divergence with large scale data, and often with neural networks, for example total correlation variational autoencoder (TC-VAE) [2], adversarial variational Bayes (AVB) [3], information maximizing GAN (InfoGAN) [4], and amortized MAP [5] all need to compute KL divergence in a deep learning setup. These large scale models have imposed new requirements on KL divergence estimation like *scalability* (able to handle large amount of data samples) and *minibatch compatibility* (compatible with minibatch-based optimization).

Methods like Nguyen et al. [1] are not suitable in the large scale setup. These modern needs were later met by modern neural network based methods such as variational divergence minimization

35th Conference on Neural Information Processing Systems (NeurIPS 2021).

(VDM) [6], mutual information neural estimation (MINE) [7], and discriminator based KL estimation with GAN-type objective [8, 5]. A key attribute of these methods is that they are based on updating a neural-net based discriminator to estimate KL divergence from a subset of samples making them scalable and minibatch compatible. We, however, noticed that even in simple examples, these methods exhibited pathologies like unreliability (high fluctuation of estimates) or instability during training (KL estimates blowing up). Similar observations of instability of VDM and MINE have also been reported in the literature [8, 9].

Why are these techniques unreliable? In this paper, we attempt to understand the core problem in the KL estimation using discriminator network. We look at it from the perspective of statistical learning theory and discriminator function space complexity and draw insights. Based on these insights, we propose that these fluctuations are a consequence of not controlling the smoothness and the complexity of the discriminator function space. Measuring and controlling the complexity of function space itself becomes a difficult problem when the discriminator is a deep neural network. Note that naive approaches to bound complexity by the number of parameters would neither be guaranteed to yield meaningful bound [10], nor be easy to implement.

Therefore, we present the following contributions to resolve these challenges. First, we propose a novel construction of the discriminator function using deep network such that it lies in a smooth function space, the Reproducing Kernel Hilbert Space (RKHS). By utilizing the learning theory and the complexity analysis of the RKHS space, we bound the probability of the error of KL-divergence estimates in terms of the radius of RKHS ball and kernel complexity. Using this bound, we propose a scalable way to control the complexity by penalizing the RKHS norm. This additional regularization of the complexity is still linear, ($O(m)$) in time complexity with the number of data samples. Then, we prove consistency of the proposed KL estimator using ideas from empirical process theory. Experimentally, we demonstrate that the proposed way of controlling complexity significantly improves KL divergence estimation and significantly reduce the variance. In mutual information estimation, our method is competitive with the state-of-the-art method and in Variational Bayesian application, our method stabilizes training of MNIST dataset leading to sharp reconstruction.

## 2 Related Work

Nguyen et al. [1] used variational method to estimate KL divergence from samples of two distribution using convex risk minimization (CRM). They used the RKHS norm as a way to both measure and penalize the complexity of the variational function. However, their work required handling all data at once and solving a convex optimization problem which has time complexity in the order of $O(m^3)$ and space complexity in the order of $O(m^2)$. Ahuja [11] used similar convex formulation in RKHS space and found it difficult to scale. VDM reformulated the f-Divergence objective using Fenchel duality and used a neural network to represent the variational function [6]. Although close in concept to [1], it is scalable since it uses a separate discriminator network and adversarial optimization. It, however, did not control the complexity of the neural-net function, and faced issues with stability.

One area of modern application of KL-divergence estimation is in computing mutual information, which is useful in applications such as stabilizing GANs [7]. MINE [7] also optimized a lower bound to KL divergence (Donsker-Varadhan representation). Similar to VDM, MINE used a neural network as the dual variational function: it is thus scalable, but without complexity control and is unstable. Another use of KL divergence is scalable variational inference (VI) as shown in AVB [8]. VI requires KL divergence estimation between the posterior and the prior, which becomes nontrivial when a sample based scalable estimation is required. AVB solved it using GAN-type adversarial formulation and a neural network discriminator. Similarly, [5] used GAN-type adversarial formulation to obtain KL divergence in amortized inference.

Chen et al. [2] proposed TC-VAE to improve disentanglement by penalizing the KL divergence between the marginal latent distribution and the product of marginals in each dimension. The KL divergence was computed by a minibatch-based sampling strategy that gives a biased estimate. Our work is close to Song et al. [9] who investigated the high variance in existing mutual information estimators and found that clipping the discriminator output is helpful in reducing variance. In our work, we take a principled way to connect variance to the complexity of discriminator function space and constrain it by penalizing its RKHS norm instead. None of the existing works considered looking at the discriminator function space, connecting its complexity to the unreliable KL-divergence estimation, or mitigating the problem by controlling the complexity.

# 3 Reproducing Kernel Hilbert Space

Let $\mathcal{H}$ be a Hilbert space of functions $f : \mathcal{X} \to \mathbb{R}$ defined on non-empty space $\mathcal{X}$. It is a Reproducing Kernel Hilbert Space (RKHS) if $\forall x \in \mathcal{X}$, the evaluation functional, $\delta_x : \mathcal{H} \to \mathbb{R}$, $\delta_x : f \mapsto f(x)$, is linear and continuous at every $f$. Every RKHS, $\mathcal{H}_K$, is associated with a unique positive definite kernel, $K : \mathcal{X} \times \mathcal{X} \to \mathbb{R}$, called the reproducing kernel [12], such that it satisfies:
1. $\forall x \in \mathcal{X}, K(.,x) \in \mathcal{H}_K$,      2. $\forall x \in \mathcal{X}, \forall f \in \mathcal{H}_K, \langle f, K(.,x) \rangle_{\mathcal{H}_K} = f(x)$

RKHS is studied using a specific integral operator. Let $\mathcal{L}_2(d\rho)$ be a space of functions $f : \mathcal{X} \to \mathbb{R}$ that are square integrable with respect to a Borel probability measure $d\rho$ on $\mathcal{X}$, we denote an integral operator $\mathscr{L}_K : \mathcal{L}_2(d\rho) \to \mathcal{L}_2(d\rho)$ [13, 14]: $(\mathscr{L}_K f)(x) = \int_{\mathcal{X}} f(y)K(x,y)d\rho(y)$. This operator will be important in constructing a function in RKHS and in computing sample complexity.

# 4 Problem Formulation and Contribution

**GAN-type Objective for KL Estimation:** Let $p(x)$ and $q(x)$ be two probability density functions in space $\mathcal{X}$ and we want to estimate their KL divergence using finite samples from each distribution in a scalable and minibatch compatible manner. As shown in [8, 5], this can be achieved by using a discriminator function. First, a discriminator $f : \mathcal{X} \to \mathbb{R}$ is trained with the objective:

$$f^* = \underset{f}{\operatorname{argmax}}[E_{p(x)} \log \sigma(f(x)) + E_{q(x)} \log(1 - \sigma(f(x)))] \tag{1}$$

where $\sigma$ is the Sigmoid function given by $\sigma(x) = \frac{e^x}{1+e^x}$. Then it can be shown [8, 5] that the KL divergence $KL(p(x)||q(x))$ is given by: $KL(p(x)||q(x)) = E_{p(x)}[f^*(x)]$

**Sources of Error:** Eq. (1) is ambiguous in the sense that it is silent about the discriminator function space over which the optimization is carried out. Typically, a neural network is used as the discriminator. This implies that we are considering the space of functions represented by the neural network of given architecture as the hypothesis space, over which the maximization occurs in eq. (1). Hence, we must rewrite eq. (1) as

$$f_h^* = \underset{f \in h}{\operatorname{argmax}}[E_{p(x)} \log \sigma(f(x)) + E_{q(x)} \log(1 - \sigma(f(x)))] \tag{2}$$

where $h$ is the discriminator function space. Furthermore, we also approximate integrals in eq. (2) with the Monte Carlo estimate using finite number of samples, say $m$, from the distribution $p$ and $q$.

$$f_h^m = \underset{f \in h}{\operatorname{argmax}}\Big[\frac{1}{m}\sum_{x_i \sim p(x_i)} \log \sigma(f(x_i)) + \frac{1}{m}\sum_{x_j \sim q(x_j)} \log(1 - \sigma(f(x_j)))\Big] \tag{3}$$

Similarly, we write KL estimate obtained from, respectively, infinite and finite samples as:

$$KL(f) = E_{p(x)}[f(x)], \quad KL_m(f) = \frac{1}{m}\sum_{x_i \sim p(x_i)}[f(x)] \tag{4}$$

Each of these steps introduce some error in our estimate. We can now start our analysis by first decomposing the total estimation error as:

$$KL_m(f_h^m) - KL(f^*) = \underbrace{KL_m(f_h^m) - KL(f_h^m)}_{\text{Deviation-from-mean error}} + \underbrace{KL(f_h^m) - KL(f_h^*)}_{\text{Discriminator induced error}} + \underbrace{KL(f_h^*) - KL(f^*)}_{Bias}$$
$$\tag{5}$$

This equation decomposes total estimation error into three terms: 1) deviation from the mean error, 2) error in KL estimate by the discriminator due to using finite samples in optimization eq. (3), and 3) bias when the considered function space does not contain the optimal function. Here, we concentrate on quantifying the probability of deviation-from-mean error which is directly related to observed variance of the KL estimate.

**Summary of Technical Contributions:** Since the deviation is the difference between a random variable and its mean, we can bound the probability of this error using concentration inequality and the complexity of the function space of $f_h^m$. To use smooth function space, we propose to construct a function out of neural networks such that it lies on RKHS (Section 5). Then, we bound the probability of deviation-from-mean error through the covering number of the RKHS space (Section 6.1), then control complexity (Section 6.2) and prove consistency of the proposed estimator (Section 7).

# 5 Constructing $f$ in RKHS

The following theorem due to Bach [15] paves a way for us to construct a neural function in RKHS.

**Theorem 1.** *[[15] Appendix A] A function $f \in \mathcal{L}_2(d\rho)$ is in Reproducing Kernel Hilbert Space, $\mathcal{H}_K$, if and only if it can be expressed as*

$$\forall x \in \mathcal{X}, f(x) = \int_{\mathcal{W}} g(w)\psi(x,w)d\tau(w), \tag{6}$$

*for a certain function $g : \mathcal{W} \to \mathbb{R}$ such that $||g||^2_{\mathcal{L}_2(d\tau)} < \infty$, $\mathcal{W}$ is a compact space and functions $w \mapsto \psi(x,w)$ are measurable for all $x$. The RKHS norm of $f$ satisfies $||f||^2_{\mathcal{H}_K} \leq ||g||^2_{\mathcal{L}_2(d\tau)}$ and the kernel $K$ is given by*

$$K(x,t) = \int_{\mathcal{W}} \psi(x,w)\psi(t,w)d\tau(w) \tag{7}$$

Theorem 1 not only gives us a condition when a square integrable function is guaranteed to lie in RKHS, it also provides us with a recipe to construct a function in RKHS. We construct $f$ using this theorem where $\psi$ and $g$ are realized with the neural networks and $d\tau$ is a probability measure with Gaussian distribution. We sample $w \sim \mathcal{N}(0, \gamma\mathrm{I})$ and pass it through two neural networks, $\psi$ and $g$, where $\psi$ takes $x$ and $w$ as two arguments and $g$ takes only $w$ as an argument. More precisely, we consider $\psi(x,w) = \phi_\theta(x)^T w$, where $\phi_\theta$ is a neural network with parameters, $\theta$. The kernel $K$, as defined in eq. (7), can be obtained as:

$$K_\theta(x^*, t^*) = \int_{\mathcal{W}} \phi_\theta(x^*)^T w w^T \phi_\theta(t^*) d\tau(w) = \gamma\phi_\theta(x^*)^T \phi_\theta(t^*) \tag{8}$$

where $E_{w \sim \mathcal{N}(0,\gamma\mathrm{I})}[ww^T] = \gamma\mathrm{I}$. We sometimes denote the kernel $K$ by $K_\theta$ to emphasize that it is a function of neural network parameters, $\theta$. Furthermore, representation of $f$ as in Theorem 1 provides us an important upper bound on the RKHS norm of $f$ as $||f||^2_{\mathcal{H}_K} \leq ||g||^2_{\mathcal{L}_2(d\tau)}$ which we will use later to bound the complexity of the discriminator function space.

Traditionally, kernel $K$ remains fixed and the norm of the function $f$ determines the complexity of the function space. In our formulation, both the RKHS kernel and the norm of $f$ with respect to the kernel changes during training since the kernel depends on neural network parameters, $\theta$. Therefore, the challenge is to tease out how neural parameters, $\theta$, affect the deviation-from-mean error in eq. (5).

# 6 Error Analysis and Control

**Assumptions:** Before starting our analysis, we list assumptions upon which our theory is based.
A1. The input domains $\mathcal{X}$ and $\mathcal{W}$ are compact.
A2. The functions $\phi_\theta$ and $g$ are Lipschitz continuous with Lipschitz constants $L_\phi$ and $L_g$ respectively.
A3. Higher order derivatives $D_x^\alpha K(x,t)$ up to some high order $\nu/2$ of kernel $K$ exist.

Assumptions A1 is satisfied in our experiments since we consider a bounded set in $\mathbb{R}^n$ and $\mathbb{R}^D$ as our domains. Similarly, A2 is satisfied since we enforce Lipschitz continuity of $\phi$ and $g$ by using spectral normalization [16]. Assumption A3 is a bit subtle. By the definition of $K$ in eq.(8), higher order derivative of $K$ exists if and only if higher order derivative of $\phi_\theta$ exists. This is readily satisfied by deep networks with smooth activation functions, and is true everywhere except at origin for ReLU activation. Using the boundedness of the input domain and Lipschitz continuity, we show the following proposition which will be useful later in the error bounds.

**Proposition 1.** *Under the assumptions A1, A2, we have $\sup_{x,t} K_\theta(x,t) < \infty$ and $||g||^2_{\mathcal{L}_2(d\tau)} < \infty$.*

## 6.1 Bounding the Error Probability of KL Estimates

Bounding the probability of deviation-from-mean error (eq. (5)) is tricky since, in our case, the kernel is not fixed and we are also optimizing over them. We bound it in two steps: 1) we derive a bound for a fixed kernel, 2) we take supremum of this bound over all the kernels parameterized by $\theta$.

For a fixed kernel, we first bound the probability of deviation-from-mean error in terms of the covering number in Lemma 1. We then use an estimate of the covering number of RKHS due to [14] to relate the bound to kernel $K_\theta$ in Theorem 2, identifying the role of neural networks in this error bound.

**Lemma 1.** *Let $f^m_{\mathcal{H}_K}$ be the optimal discriminator function in an RKHS $\mathcal{H}_K$ which is bounded by $M$ with respect to $||.||_\infty$. Let $KL_m(f^m_{\mathcal{H}_K}) = \frac{1}{m}\sum_i f^m_{\mathcal{H}_K}(x_i)$ and $KL(f^m_{\mathcal{H}_K}) = E_{p(x)}[f^m_{\mathcal{H}_K}(x)]$ be the estimate of KL divergence from m samples and that by using the true distribution $p(x)$ respectively. Then the probability of error at some accuracy level, $\epsilon$, is lower-bounded as:*

$$Prob.(|KL_m(f^m_{\mathcal{H}_K}) - KL(f^m_{\mathcal{H}_K})| \le \epsilon) \ge 1 - 2\mathcal{N}(\mathcal{H}_K, \frac{\epsilon}{4\sqrt{S_K}}) \exp(-\frac{m\epsilon^2}{4M^2})$$

*where $\mathcal{N}(\mathcal{H}_K, \eta)$ denotes the covering number of an RKHS space $\mathcal{H}_K$ with disks of radius $\eta$, and $S_K = \sup_{x,t} K(x,t)$ which we refer to as kernel complexity.*

*Proof Sketch.* We cover RKHS with discs of radius $\eta = \frac{\epsilon}{4\sqrt{S_K}}$. Within this radius, the deviation does not change too much. So, we can bound deviation probability at the center of disc and apply union bound over all the discs. To bound deviation probability at the center, we apply Hoeffding's inequality and applying union bound simply leads to counting number of discs which is exactly the covering number. See supplementary materials for the full proof. □

Lemma 1 bounds the probability of error in terms of the covering number of the RKHS space. Note that the radius of the disc is inversely related to $S_K$ which indicates how complex the RKHS space defined by the kernel $K_\theta$ is. Here $K_\theta$ depends on the neural network parameters $\theta$. Therefore, we denote $S_K$ as a function of $\theta$ as $S_K(\theta)$ and term it kernel complexity. Next, we use Lemma 2 due to [14] to obtain an error bound in estimating KL divergence with finite samples in Theorem 2.

**Lemma 2** ([14]). *Let $K : \mathcal{X} \times \mathcal{X} \to \mathbb{R}$ be a $\mathcal{C}^\infty$ Mercer kernel and the inclusion $I_K : \mathcal{H}_K \hookrightarrow \mathcal{C}(\mathcal{X})$ be the compact embedding defined by $K$ to the Banach space $\mathcal{C}(\mathcal{X})$. Let $B_R$ be the ball of radius $R$ in RKHS $\mathcal{H}_K$. Then $\forall \eta > 0, R > 0, \nu > n$, we have*

$$\ln \mathcal{N}(I_K(B_R), \eta) \le \left(\frac{RC_\nu}{\eta}\right)^{\frac{2n}{\nu}} \tag{9}$$

*where $\mathcal{N}$ gives the covering number of the space $I_K(B_R)$ with discs of radius $\eta$, and $n$ represents the dimension of the input space $\mathcal{X}$. $C_\nu$ is given by $C_\nu = C_s\sqrt{||\mathscr{L}_s||}$ where $\mathscr{L}_s$ is a linear embedding from square integrable space $\mathcal{L}_2(d\rho)$ to the Sobolev space $H^{\nu/2}$, $||\mathscr{L}_s||$ denotes operator norm and $C_s$ is a constant.*

To prove Lemma 2 [14], the RKHS space is embedded in the Sobolev Space $H^{\nu/2}$ using $\mathscr{L}_s$ and then the covering number of the Sobolev space is used. Thus the norm of $\mathscr{L}_s$ and the degree of Sobolev space, $\nu/2$, appears in the covering number of a ball in $\mathcal{H}_K$. In Theorem 2, we use Lemma 1 and 2 to bound the estimation error of KL divergence.

**Theorem 2.** *Let $KL(f^m_{\mathcal{H}})$ and $KL_m(f^m_{\mathcal{H}})$ be the estimates of KL divergence obtained by using true distribution $p(x)$ and $m$ samples respectively as described in Lemma 1, then the probability of error in the estimation at the error level $\epsilon$ is given by:*

$$Prob.(|KL_m(f^m_{\mathcal{H}}) - KL(f^m_{\mathcal{H}})| \le \epsilon) \ge 1 - 2\exp\left[\left(\frac{4RC_p\sqrt{S_p||\mathscr{L}_p||}}{\epsilon}\right)^{\frac{2n}{\nu}} - \frac{m\epsilon^2}{4M^2}\right]$$

*where $C_p\sqrt{S_p||\mathscr{L}_p||} = \sup_{K_\theta} C_s\sqrt{S_K(\theta)||\mathscr{L}_s||}$, i.e. $C_p, S_p, \mathscr{L}_p$ correspond to a kernel for which the bound is maximum.*

*Proof.* We prove this in two steps: First we obtain an error bound for a fixed kernel space and apply supremum over all $\theta$. For any RKHS $\mathcal{H}_{K_\theta}$, with fixed kernel $K_\theta$, we have

$$\text{Prob.}(|KL_m(f^m_{\mathcal{H}_{K_\theta}}) - KL(f^m_{\mathcal{H}_{K_\theta}})| \ge \epsilon) \le 2\exp\left[\left(\frac{4RC_s\sqrt{S_K(\theta)||\mathscr{L}_s||}}{\epsilon}\right)^{\frac{2n}{\nu}} - \frac{m\epsilon^2}{4M^2}\right] \tag{10}$$

We prove this error bound as follows. Lemma 2 gives the covering number of an RKHS ball of radius $R$, which we apply to Lemma 1. We fix the radius of discs to $\eta = \frac{\epsilon}{4\sqrt{S_K}}$ in Lemma 1 and substitute $C_\nu = C_s\sqrt{||\mathscr{L}_s(\theta)||}$ to obtain eq.(10).

Since we are continuously changing $\theta$ during training, the kernel also changes. Hence, to find the upper bound over all possible kernels, we take the supremum over all kernels.

$$\text{Prob.}(|KL_m(f^m_{\mathcal{H}}) - KL(f^m_{\mathcal{H}})| \geq \epsilon) \leq \sup_{K_\theta} \text{ Prob.}(|KL_m(f^m_{\mathcal{H}_{K_\theta}}) - KL(f^m_{\mathcal{H}_{K_\theta}})| \geq \epsilon) \quad (11)$$

$$\leq 2\exp\left[\left(\frac{4RC_p\sqrt{S_p||\mathscr{L}_p||}}{\epsilon}\right)^{\frac{2n}{\nu}} - \frac{m\epsilon^2}{4M^2}\right] \quad (12)$$

where $S_p = S_K(\theta_p)$ and $\mathscr{L}_p = \mathscr{L}_K(\theta_p)$, *i.e.*, $S_p$ and $\mathscr{L}_p$ correspond to kernel complexity and Sobolev operator norm corresponding to optimal kernel $K_{\theta_p}$ that extremizes eq. (11). Theorem statement readily follows from eq. (12) □

Theorem 2 shows that the error increases exponentially with the radius of the RKHS space, $R$, complexity of the kernel $S_K(\theta_p)$, and the norm of the Sobolev space embedding operator $||\mathscr{L}_p||$. The Sobolev embedding operator, $\mathscr{L}_p$, is a mapping from $\mathcal{L}_2(d\rho)$ to the Sobolev space $H^{\nu/2}$. It can be shown [14] that the operator norm can be bounded as $||\mathscr{L}_p|| \leq \rho(\mathcal{X})\sum_{|\alpha|\leq\nu/2}\sup_{x,t\in\mathcal{X}}(D_x^\alpha K_{\theta_p}(x,t))^2$, where $\rho$ is the measure of the input space $\mathcal{X}$. Therefore, the norm $||\mathscr{L}_p||$ directly measures smoothness of $K_{\theta_p}$ while $S_K(\theta_p)$ only depends on the supremum value of $K_{\theta_p}$. $C_p$ is a constant not depending on $S_p$, $\mathscr{L}_p$ or $R$, and it is always finite (see [14, 17] for more details). $S_p$ and $\mathscr{L}_p$ are related to $\phi, \gamma$ through the kernel $K_{\theta_p}$. $S_p$ is defined as $S_p = \sup_{x,t} K(x,t) = \sup_{x,t} \gamma\phi_{\theta_p}^T(x)\phi_{\theta_p}(t)$, so it is directly related to the network $\phi_{\theta_p}$. We show that this is finite due to Lipschitz constraint on $\phi_{\theta_p}$ as described in Assumption A2 (see Supplementary material for proof). $\mathscr{L}_p$ is upper bounded as follows: $||\mathscr{L}_p|| \leq \rho(\mathcal{X})\sum_{|\alpha|\leq\nu/2}\sup_{x,t\in\mathcal{X}}(D_x^\alpha K_{\theta_p}(x,t))^2 = \rho(\mathcal{X})\sum_{|\alpha|\leq\nu/2}\sup_{x,t\in\mathcal{X}}(D_x^\alpha\gamma\phi_{\theta_p}^T(x)\phi_{\theta_p}(t))^2$ where $D$ is a differentiation operator. Therefore, $\mathscr{L}_p$ is finite if the higher order derivatives of $\phi_{\theta_p}$ exist and are finite.

## 6.2 Complexity Control

From Theorem 2, we see that the error probability could be decreased by decreasing $R, ||\mathscr{L}_p||$ and $S_K(\theta_p)$. Using argument similar to the proof of Proposition 1, we can show that the Lipschitz constraint on $\phi_\theta$ also affects $S_K$ and may affect $||\mathscr{L}_p||$. In our experiments, however, we fix the Lipschitz constraints during optimization and do not change $S_K$ and $||\mathscr{L}_p||$ dynamically. Here, we focus on the norm, $R$ from Theorem 2. From Theorem1, we know that the RKHS norm is upper bounded by the norm of function $g$ as $||f||^2_{\mathcal{H}_K} \leq ||g||^2_{\mathcal{L}_2(d\tau)}$. We have access to finite approximation of $||g||^2_{\mathcal{L}_2(d\tau)}$ by construction and hence, we can use it to penalize the complexity during optimization. To obtain the optimal discriminator $f^m_h$, we optimize the following objective with an extra penalization of the upper bound, i.e. $||g||_{\mathcal{L}_2(d\tau)}$ on the RKHS norm of $f$:

$$f^m_h = \underset{f\in h}{\text{argmax}}\frac{1}{m}\sum_{x_i\sim p(x_i)}\log\sigma(f(x_i)) + \frac{1}{m}\sum_{x_j\sim q(x_j)}\log(1-\sigma(f(x_j))) - \frac{\lambda_0}{m}||g||^2_{\mathcal{L}_2(d\tau)} \quad (13)$$

The regularization term prevents the radius of RKHS ball from growing, maintaining a low error probability. Optimization of eq. (13) w.r.t. neural network parameters $\theta$ allows dynamic control of the complexity of the discriminator function on the fly in a scalable and efficient way. Note that, computation of $||g||_{\mathcal{L}_2(d\tau)}$ requires randomly sampling $w \sim \mathcal{N}(0, \gamma\mathbf{I})$ and passing through neural network $g$ independent of the data $x_i, x_j$. *Therefore, if the computational complexity of optimization is O(m), it will remain the same after incorporating this additional term, i.e. regularization does not increase asymptotic time complexity which is linear with the number of samples, m.*

## 7 Variance and Consistency of the Estimate

### 7.1 Variance Analysis

Theorem 2 gives an upper bound on the probability of error. Intuitively, the variance and probability of error behave similarly for many distributions, i.e. higher variance might indicate higher probability of error. Below we quantify this intuition for a Gaussian distributed estimate:

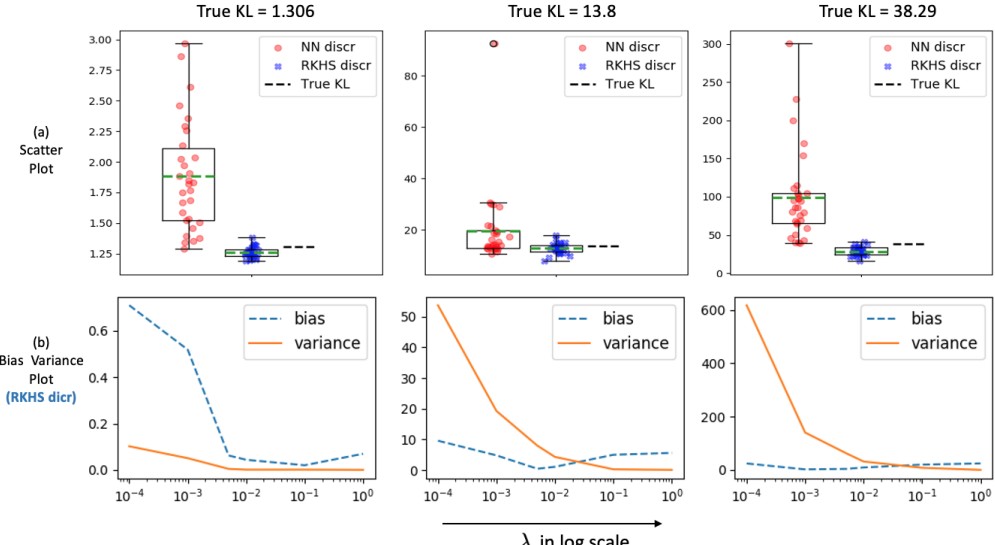

Figure 1: a) Top scatter plot compares KL divergence estimates between a method using Neural network discriminator without complexity control (red) and that using RKHS discriminator with compelxity control (blue); b) In the bottom, we show the effect of varying the regularization parameter $\lambda$ on bias and variance while using the RKHS discriminator with complexity control as in eq.(13).

**Theorem 3.** *Let $X = KL_m(f_{\mathcal{H}}^m)$ be the estimated KL divergence using m samples as described in Theorem 2. Assuming that X follows a Gaussian distribution $X \sim \mathcal{N}(\mu, \varsigma^2)$, we can obtain an upper bound on the standard deviation of the estimate as follows:*

$$\varsigma \leq \frac{\epsilon}{erf^{-1}\Big[ -4\exp\Big[ \Big( \frac{4RC_p\sqrt{S_p||\mathscr{L}_p||}}{\epsilon} \Big)^{\frac{2n}{\nu}} - \frac{m\epsilon^2}{4M^2} \Big] + 1 \Big]}$$

*where erf is the Gauss error function and is a monotonic function.*

Theorem 3 suggests that by decreasing $R$, the radius of the RKHS ball, the variance of the estimate could be decreased. Experimentally, we observe that the variance decreases as we penalize the RKHS norm more, consistent with the spirit of Theorem 3.

Note that Theorem 3 makes a strong assumption that the estimate is distributed as a Gaussian distribution. While it gives us good intuition about the decay of variance as the complexity increases, it is natural to inquire about the validity of this type of relation in a more general sense without assumption on the probability distribution of the estimate. To make a general statement, the key idea is to understand how the cumulative distribution function (CDF) is related to the variance. To clarify this point further, let's look at the eq.(33) in the proof of Theorem 3 in supplementary material: $1 - \Phi_{\mu,\varsigma}(\mu + \epsilon) \leq 2\exp\left[ \left( \frac{4RC_s\sqrt{S_K||\mathscr{L}_s||}}{\epsilon} \right)^{\frac{2n}{\nu}} - \frac{m\epsilon^2}{4M^2} \right]$. This equation connects the CDF to the variables like $S_K$, $R$, $\mathscr{L}_s$ of the discriminator function space without assuming anything about the shape of the distribution. For a Gaussian distribution, we plug in CDF of a Gaussian distribution, $\Phi_{\mu,\varsigma}(\hat{x}) = \frac{1}{2}\left[ 1 + erf\left( \frac{\hat{x}-\mu}{\varsigma\sqrt{2}} \right) \right]$ and obtain the result of Theorem 3. For any other distribution, we can carry out similar analysis. A key factor that determines the behavior between the variance and the discriminator complexity is how variance appears in the CDF expression. For example, in both Gaussian distribution and in exponential distribution, we know that the relation between CDF function and variance is inversely related. We believe that as long as this inverse type of relation between CDF and variance holds, we can obtain statements like Theorem 3 for other distributions as well. This leads to a key insight: *similar to the Gaussian case, the decaying behavior of the variance with decreasing complexity holds as long as the estimate is distributed such that its CDF function has inverse relation with the variance.*

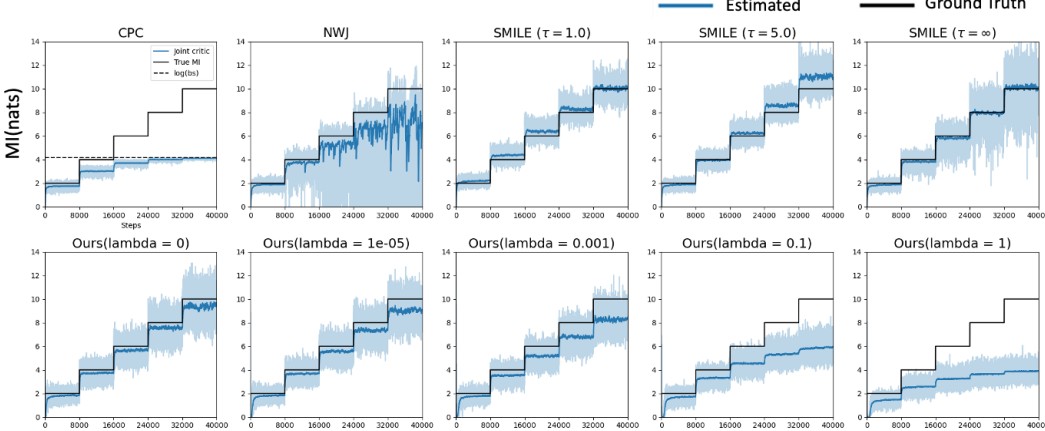

Figure 2: Comparing our method with CPC [19], convex risk minimization(NWJ) [1] and SMILE [9] regarding mutual information estimation between two variables.

## 7.2 Consistency of Estimates

In eq.(13), we use a regularized objective to obtain an optimal discriminator instead of an unregularized objective as in eq.(3). It is important to show that the estimator of KL divergence obtained by using this regularized objective is consistent and approaches the true estimate in the limiting case as the number of data samples grows to infinity. In the following theorem, we show this.

**Theorem 4.** *Let $f^*$ and $f^m$ be optimal discriminators as described in eq. (1) and eq. (13) respectively, and the KL estimate is given by $KL(f) = E_{p(x)}[f(x)], \quad KL_m(f) = \frac{1}{m}\sum_{x_i \sim p(x_i)}[f(x)]$. Then, in the limiting case as $m \to \infty$, $|KL_m(f_h^m) - KL(f^*)| \to 0$ in probability.*

*Proof Sketch.* The difference between the true KL divergence and the estimated KL divergence can be divided into three terms as shown in eq. (5). We assume that our function space is rich enough to contain the true solution, driving bias to zero. From Theorem 2, we see that in the limiting case of $m \to 0$, the deviation-from-mean error goes to 0. Therefore, the key step that remains to be shown is that the discriminator induced error (second term in eq.(5)) also goes to 0 as $m \to \infty$.

It can be shown if we can prove that the optimal discriminator in eq. (13) approaches the optimal discriminator in eq. (2). To prove this, we show that the argument being maximized by $f_h^m$ approaches the argument being maximized by $f_h^*$ in the limiting case. To show this, we need to show that the function space, $\log \sigma f$, is Glivenko Cantelli [18], which we prove in following steps:
1. We show that $f$ is Lipschitz continuous by definition and due to Lipschitz continuity of $\phi_\theta$. Then we show that $\log \sigma f$ is Lipschitz continuous if $f$ is Lipschitz continuous.
2. Then we show that for a class of functions with Lipschitz constant $L$, the metric entropy, $\log N$, can be obtained in terms of $L$ and entropy number of the bounded input space, $\mathcal{X}$.
3. Since the metric entropy does not grow with the number of samples $m$, we show that $\frac{1}{m}\log N \to 0$ which lets us show that $\log \sigma f$ belongs to Glivenko Cantelli class of functions by using Theorem 2.4.3 from [18]. See supplementary material for the complete proof. □

## 8 Experimental Results

We present results on three applications of KL divergence estimation: 1. KL estimation between simple Gaussian distributions, 2. Mutual information estimation, 3. Variational Bayes. In our experiments, the RKHS discriminator is constructed with $\psi$ and $g$ networks as described in Section 5, where the network $\psi$ is very close to a regular neural network. In two experiments, we compare our results with the models using regular neural net discriminator to ensure that the difference in performance between RKHS and regular neural network is not due to architectural difference. Our code is publicly available at https://github.com/sandeshgh/Reliable-KL-estimation

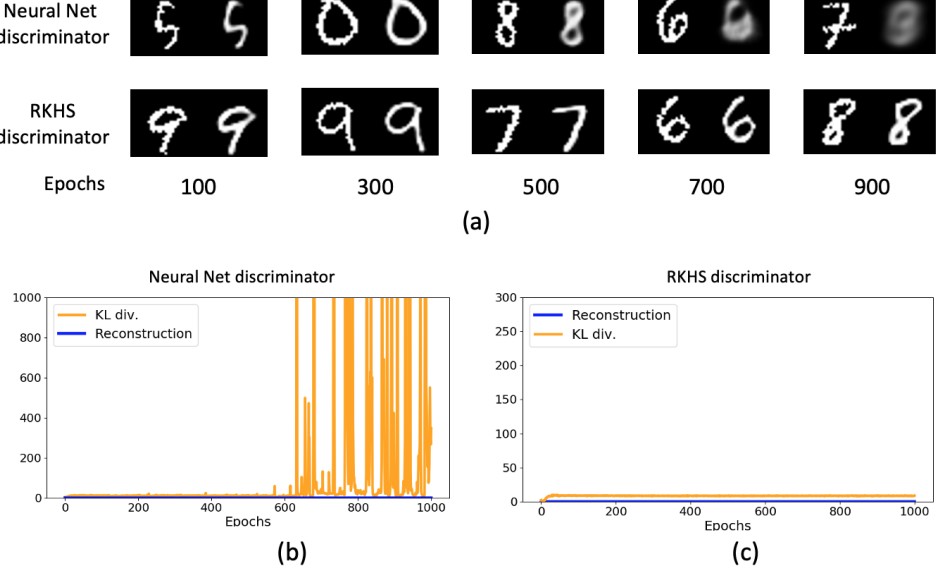

Figure 3: (a) Comparison of MNIST digit reconstruction using AVB autoencoder model [8]. Trace of KL divergence and reconstruction loss in AVB model with Neural network discriminator (b) and RKHS discriminator in (c).

**KL Estimation between Two Gaussians**   We assume that we have finite sets of samples from two distributions. We further assume that we are required to apply minibatch based optimization. We consider estimating KL divergence between two Gaussian distributions in 2D, where we know the analytical KL divergence between the two distributions as the ground truth. We consider three different pairs of distributions corresponding to true KL divergence values of $1.3, 13.8$ and $38.29$, respectively and use $m = 5000$ samples from each distribution to estimate KL in the finite case. We repeat the estimation experiments with random initialization 30 times and report the mean, standard deviation, scatter and box plots.

Fig. 1 top row compares the estimation of KL divergence with regular neural net and RKHS discriminator with complexity control based on eq. (13). With our proposed RKHS discriminator, the KL estimates are significantly more reliable and accurate: error reduced from 0.5 to 0.04, 5.8 to 1.07 and 60.6 to 9.7 and variance reduced from 0.2 to 0.002, 223 to 4.4 and 3521 to 33 for true KL 1.3, 13.8 and 38.29 respectively. In Fig. 1 bottom row, we investigate our complexity control method on the effect of varying the regularization parameter $\lambda = \lambda_0/m$. As expected, increasing regularization parameter penalizes more on the RKHS norm and therefore reduces variance. This is consistent with our theory. Regarding bias, however, as we increase the $\lambda$, the bias decreases and then starts to increase. Hence, one needs to strike a balance between bias and variance while choosing $\lambda$.

**Mutual Information Estimation**   Computation of mutual information is a direct use case of KL divergence computation. We replicate the experimental setup of [20, 9] to estimate mutual information between $(x, y)$ drawn from 20-d Gaussian distributions, where the mutual information is increased by step size of 2 from 2 to 10. We compare the performance of our method with traditional KL divergence computation methods like contrastive predictive coding (CPC) [19], convex risk minimization (NWJ) [1] and SMILE [9]. In Fig.2, our method with RKHS discriminator (with $\lambda = 1e^{-5}$) performs better than CPC [19] and NWJ [1], and is competitive with the state-of-the-art, SMILE [9]. In the bottom row, we also show the effect of regularization parameter $\lambda$ in our method. Similar to the previous experiment, increasing the regularization parameter decreases the variance and increases the bias. It is consistent with our theoretical insights about the effect of reducing RKHS norm on variance.

**Adversarial Variational Bayes**   Variational Bayes requires KL divergence estimation. When we do not have access to analytical form of the posterior/prior distributions, but only have access to the samples, we need to estimate KL divergence from samples. Adversarial Variational Bayes (AVB) [8] presents a way to achieve this using a discriminator network. We adopt this setup and demonstrate that the training becomes unstable if we do not constrain the complexity of the discriminator. First, we

| Epoch | 100 | 200 | 300 | 400 | 500 | 600 | 700 | 800 | 900 |
|-------|------|------|------|------|------|------|------|------|------|
| NNet | 49.64 | 43.25 | 47.52 | 44.29 | 51.31 | 52.45 | 148.9 | 157.5 | 261.9 |
| RKHS | 37.58 | 33.18 | 31.46 | 31.39 | 30.36 | 29.37 | 28.17 | 28.18 | 28.17 |

Table 1: FID score (smaller the better) at different epochs of training between the reconstruction (using respective discriminator in VAE) and the ground truth.

train AVB on MNIST dataset with a simple neural network discriminator architecture. As the training progresses, the KL divergence blows up after about 500 epochs (Fig. 3(b)) and the reconstruction starts to get worse (Fig. 3(a)). We modify the same architecture according to our construction such that the discriminator lies in the RKHS and then penalize the RKHS norm as in eq. (13). It stabilizes the training for a large number of epochs as shown in Fig. 3(c) and the reconstruction does not deteriorate as the training progresses, resulting into sharp reconstruction (Fig. 3(a)). To make this comparison more precise and quantitative, we compute FID score between reconstruction and the ground truth after each epoch as tabulated in Table 1. In the earlier epochs, the FID score ( and the reconstruction in Fig.3(a)) is okay even for the neural net discriminator, However, this score is worse than using the RKHS discriminator, where the scores are in the range of 30-40 upto epoch 400. For the neural network discriminator, the score increases (worsens) in the mid epochs and becomes completely unstable with FID score shooting up to 262 at epoch 900. On the other hand, for the RKHS discriminator, the score steadily and smoothly decreases as the epoch increases reaching the best 28.17 at epoch 900. This experiment demonstrates that the proposed RKHS discriminator with norm regularization is both reliable and effective in terms of standard metric like FID.

We want to clarify that this instability in training neural net discriminator is present if we use a basic discriminator architecture. It does not mean that there exists no other method to design a stable neural net discriminator. In fact, AVB [8] presents a discriminator that adds additional inner product structure to stabilize the discriminator training. Our point here is that we can stabilize the training by ensuring that the discriminator lies in a well behaved function space (the RKHS) and controlling its complexity, consistent with our theory.

## 9 Limitations, Discussion and Conclusion

**Limitations:** The proposed construction of neural function in RKHS exhibits good properties of both the deep learning and kernel methods. However, it requires constructing two separate deep networks, $\psi$ and $g$. It makes our model a bit bulky and also requires more parameter due to additional $g$. Moreover, currently our RKHS discriminator's output is scalar; generalizing this function to a multivariate output could make our model bulkier and increase parameters even more. Second limitation is the requirement of higher order derivative of kernel $K$ in assumption A3. While this requirement is satisfied if smooth activation function is used in $\phi_\theta$, for activations like ReLU or LeakyReLU, the derivatives exist everywhere except at the origin. In these cases, we need to carefully investigate if we can use subgradients to define operator norm $||\mathscr{L}_p||$.

**Discussion and Conclusion:** We have shown that using a regular neural network as a discriminator in estimating KL divergence results in unreliable estimation if the complexity of the function space is not controlled. We then showed a solution by constructing a discriminator function in RKHS space using neural networks and penalizing its complexity in a scalable way. Although the idea to use RKHS norm to penalize complexity is not new (see for example [1]), it is not clear how to use this idea directly on the function $f$. In traditional kernel methods, algorithms often do not work with RKHS function $f$ directly, but rather work with kernel matrix, $K$ by using, for example, the Representer Theorem [21]. In the case of big data, working with the big kernel matrix is computationally expensive although some methods have been proposed to speed up the computation, like Random Fourier Feature [22]. We propose a different view by directly constructing a function in RKHS space, which led us to scalable algorithm while incorporating the advantages of neural networks. Moreover, our representation could also be seen as an improvement over RFF by using neural basis, $\psi$, instead of Fourier basis. The idea of constructing a neural-net function in RKHS and complexity control could also be useful in stabilizing GANs in general. Currently, the most successful way to stabilize GANs is to enforce smoothness by gradient penalization [23, 24, 25]. On the light of the present analysis, gradient penalty could also be thought as a way to control the complexity of the discriminator.

## 10    Acknowledgements and Disclosure of Funding

We would like to express our deep gratitude to Prof. Linwei Wang and Dr. Prashnna Kumar Gyawali for the helpful discussion at the initial stage of this work. Their discussion was especially important in the inception of this project, in identifying the need for KL estimation from samples and limitation of existing KL estimation approaches. We would like to thank Prof. Dana H. Brooks for the helpful discussions and giving exposure to this work in the early stage. We are grateful to Prof. Octavia Camps for her support, encouragements and resources. We are very thankful to Zulqarnain Khan for his feedback on the early draft of the paper.

Funding disclosure: We are thankful to the support from NIH/NCI R01CA240771 and NIH/NHLBI U01HL089856 .

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
