# Reliable Estimation of KL Divergence using a Discriminator in Reproducing Kernel Hilbert Space Supplementary Material

**Sandesh Ghimire** [*]**, Aria Masoomi, Jennifer Dy**
Department of Electrical and Computer Engineering
Northeastern University
sandesh@ece.neu.edu, masoomi.a@northeastern.edu, jdy@ece.neu.edu

**Organization:**   This supplementary material is presented in a format parallel to the main paper. The section numbers and titles are consistent with the main paper. But, here we also add one new section: Section 10 where we describe the societal impacts and possible negative impacts of the paper. Similarly, the Theorem numbers are consistent with the main paper, but we also have several additional theorems and lemmas which were not included in the main paper.

## 4   Problem Formulation and Contribution

**GAN-type Objective for KL Estimation**   Let $f$ be a discriminator, $f : \mathcal{X} \to \mathbb{R}$. Let $p(x)$ and $q(x)$ be two probability density functions defined over the space $\mathcal{X}$. First, we train a discriminator as:

$$f^* = \underset{f}{\operatorname{argmax}}[E_{p(x)} \log \sigma(f(x)) + E_{q(x)} \log(1 - \sigma(f(x)))] \tag{1}$$

where $\sigma$ is the Sigmoid function given by $\sigma(x) = \frac{e^x}{1+e^x}$. Then the KL divergence $KL(p(x)||q(x))$ is given by:

$$KL(p(x)||q(x)) = E_{p(x)}[f^*(x)] \tag{2}$$

*Proof.*  The proof is based on similar proofs in [7, 8] and presented here for the sake of completeness.

We rewrite the objective as :

$$\int p(x) \log \sigma(f(x)) + q(x) \log(1 - \sigma(f(x)))dx \tag{3}$$

This integral is maximum with respect to $f$ if and only if the integrand is maximal for every $x$. As argued in the Proposition 1 of [2], the function

$$t \mapsto a \log(t) + b \log(1 - t) \tag{4}$$

attains its maximum at $t = \frac{a}{a+b}$ showing that,

$$\sigma(f^*(x)) = \frac{p(x)}{p(x) + q(x)} \tag{5}$$

Plugging the expression for Sigmoid function, we obtain,

$$f^*(x) = \frac{p(x)}{q(x)} \tag{6}$$

Therefore, by the definition of KL divergence, we have:

$$\mathrm{KL}(p(x)||q(x)) = E_{p(x)}[\frac{p(x)}{q(x)}] = E_{p(x)}[f^*(x)] \tag{7}$$

$\square$

---

[*]Webpage: https://sandeshgh.com/

35th Conference on Neural Information Processing Systems (NeurIPS 2021), Sydney, Australia.

# 6 Error Analysis and Control

We start with the set of assumptions based on which our theory is developed.

A1. The input domains $\mathcal{X}$ and $\mathcal{W}$ are compact.

A2. The functions $\phi_\theta$ and $g$ are Lipschitz continuous with Lipschitz constant $L_\phi$ and $L_g$ respectively.

A3. Higher order derivatives $D_x^\alpha K(x, t)$ of kernel $K$ exist up to some high order $\nu/2$ .

**Proposition 1.** *Under the assumptions A1, A2, we have*
*i)* $\sup\limits_{x,t} K_\theta(x, t) < \infty$, *and*
*ii)* $||g||^2_{\mathcal{L}_2(d\tau)} < \infty$.

*Proof.* i) By the definition $K_\theta(x, t) = \gamma\langle\phi_\theta(x), \phi_\theta(t)\rangle$. Using Cauchy Schwartz,

$$K_\theta(x, t) \leq \gamma||\phi_\theta(x)||\,||\phi_\theta(t)|| \tag{8}$$
$$\leq \gamma L_\phi||x||L_\phi||t|| \tag{9}$$
$$< \infty \tag{10}$$

where we used the fact that $\mathcal{X}$ is bounded, and therefore, $||x||$ and $||t||$ are finite.
ii) By definition,

$$||g||^2_{\mathcal{L}_2(d\tau)} = \int g(w)^2 d\tau(w) \tag{11}$$

$$\leq \int L_g^2||w||^2 d\tau(w) \tag{12}$$

$$= L_g^2 tr(C_w) \tag{13}$$

where $C_w$ is the uncentered covariance matrix of the Gaussian distributed w. Therefore, we immediately obtain $||g||^2_{\mathcal{L}_2(d\tau)} < \infty$. $\qquad\square$

These results are useful in constructing a function $f$ in RKHS in Theorem 1 (Section 5) of the main paper.

## 6.1 Bounding the Error Probability of KL Estimates

We bound the deviation-from-mean error in two steps: 1) we derive a bound for a fixed kernel, 2) we take supremum of this bound over all the kernels parameterized by $\theta$.

For a fixed kernel, we first bound the probability of deviation-from-mean error in terms of the covering number in Lemma 1. Then, we use an estimate of the covering number of RKHS due to [1] to obtain a bound of error probability in terms of the kernel $K_\theta$ in Lemma 3. Note that, Lemma 3 is proved for a fixed kernel $K_\theta$, where $\theta$ is fixed. Then finally in Theorem 2, we take supremum over all kernels $K_\theta$s to obtain a bound on error probability on a space of functions with all possible kernels.

**Lemma 1.** *Let $f^m_{\mathcal{H}_K}$ be the optimal discriminator function in an RKHS $\mathcal{H}_K$ which is bounded by $M$ with respect to $||.||_\infty$. Let $KL_m(f^m_{\mathcal{H}_K}) = \frac{1}{m}\sum_i f^m_{\mathcal{H}_K}(x_i)$ and $KL(f^m_{\mathcal{H}_K}) = E_{p(x)}[f^m_{\mathcal{H}_K}(x)]$ be the estimate of KL divergence from m samples and that by using true distribution $p(x)$ respectively. Then the probability of error at some accuracy level, $\epsilon$ is lower-bounded as:*

$$Prob.(|KL_m(f^m_{\mathcal{H}_K}) - KL(f^m_{\mathcal{H}_K})| \leq \epsilon) \geq 1 - 2\mathcal{N}(\mathcal{H}_K, \frac{\epsilon}{4\sqrt{S_K}})\exp(-\frac{m\epsilon^2}{4M^2})$$

*where $\mathcal{N}(\mathcal{H}_K, \eta)$ denotes the covering number of a RKHS space $\mathcal{H}_K$ with disks of radius $\eta$, and $S_K = \sup\limits_{x,t} K(x, t)$ which we refer as kernel complexity*

*Proof.* Let $\ell_z(f) = E_{p(x)}[f(x)] - \frac{1}{m}\sum_i f(x_i)$ denotes the error in the estimate such that we want to bound $|\ell_z(f)|$. We have,

$$\ell_z(f_1) - \ell_z(f_2) = E_{p(x)}[f_1(x) - f_2(x)] - \frac{1}{m}\sum_i f_1(x_i) - f_2(x_i)$$

We know $E_{p(x)}[f_1(x) - f_2(x)] \le ||f_1 - f_2||_\infty$ and $\frac{1}{m}\sum_i f_1(x_i) - f_2(x_i) \le ||f_1 - f_2||_\infty$. Using the triangle inequality, we obtain $|\ell_z(f_1) - \ell_z(f_2)| \le 2||f_1 - f_2||_\infty$. Now, consider $f \in \mathcal{H}_K$, then,

$$|f(x)| = |\langle K_x, f\rangle| \le ||f||||K_x|| = ||f||\sqrt{K(x,x)} \tag{14}$$

This implies the RKHS space norm and $\ell_\infty$ norm of a function are related by

$$||f||_\infty \le \sqrt{S_K}||f||_{\mathcal{H}_K} \tag{15}$$

Hence, we have:

$$|\ell_z(f_1) - \ell_z(f_2)| \le 2\sqrt{S_K}||f_1 - f_2||_{\mathcal{H}_K} \tag{16}$$

The idea of the covering number is to cover the whole RKHS space $\mathcal{H}_K$ with disks of some fixed radius $\eta$, which helps us bound the error probability in terms of the number of such disks. Let $\mathcal{N}(\mathcal{H}_K, \eta)$ be such disks covering the whole RKHS space. Then, for any function $f$ in $\mathcal{H}_K$, we can find some disk, $D_j$ with centre $f_j$, such that $||f - f_j||_{\mathcal{H}_K} \le \eta$. From eq.(16), we have $|\ell_z(f) - \ell_z(f_j)| \le 2\sqrt{S_K}||f - f_j||_{\mathcal{H}_K} \le 2\sqrt{S_K}\eta$. Now, we choose $\eta = \frac{\epsilon}{2\sqrt{S_K}}$. Plugging in $\epsilon = 2\sqrt{S_K}\eta$ in the previous equation, we obtain, $|\ell_z(f) - \ell_z(f_j)| \le \epsilon$. This is true for any $f$ within the disk $D_j$. Now, we show $\ell_z(f_j) \ge |\ell_z(f)| - \epsilon$. To show this, $|\ell_z(f)| = |\ell_z(f) - \ell_z(f_j) + \ell_z(f_j)| \le |\ell_z(f) - \ell_z(f_j)| + |\ell_z(f_j)|$ using triangle inequality. Therefore, $|\ell_z(f)| \le \epsilon + |\ell_z(f_j)|$, from which we obtain $\ell_z(f_j) \ge |\ell_z(f)| - \epsilon$. From this statement, it is easy to see that, for any $f \in D_j$, if $|\ell_z(f)| \ge 2\epsilon$, then it must be true that $|\ell_z(f_j)| \ge \epsilon$. Concisely, we write

$$\sup_{f \in D_j} |\ell_z(f)| \ge 2\epsilon \implies |\ell_z(f_j)| \ge \epsilon \tag{17}$$

Using the Hoeffding's inequality, $\text{Prob.}(|\ell_z(f_j)| \ge \epsilon) \le 2e^{-\frac{m\epsilon^2}{2M^2}}$ and eq.(17), we can show

$$\text{Prob.}(\sup_{f \in D_j} |\ell_z(f)| \ge 2\epsilon) \le 2e^{-\frac{m\epsilon^2}{2M^2}} \tag{18}$$

The reasoning behind this equation is as follows. Let us denote the left hand side and right hand side of eq.(17) as events A and B; i.e., $\sup_{f \in D_j}|\ell_z(f)| \ge 2\epsilon$ is denoted by $A$ and $|\ell_z(f_j)| \ge \epsilon$ as B. Then eq.(17) says $A$ implies $B$. That means whenever A happens, B also happens, but not vice versa. Therefore, represented as a set, set A is possibly smaller than B, and furthermore, A is a subset of B. When A is a subset of B, we have $Pr(A) \le Pr(B)$ by using the Monotonicity property of probability measure. Using Hoeffding's inequality, we can further obtain, $Pr(B) = Pr(|\ell_z(f_j)| \ge \epsilon) \le 2e^{\frac{-m\epsilon^2}{2M^2}}$. Since $Pr(A) \le Pr(B)$, we obtain, $Pr(A) \le 2e^{\frac{-m\epsilon^2}{2M^2}}$ which is exactly eq.(18).

Applying union bound over all the disks, we obtain,

$$\text{Prob.}(\sup_{f \in \mathcal{H}}|\ell_z(f)| \ge 2\epsilon) \le 2\mathcal{N}(\mathcal{H}, \frac{\epsilon}{2\sqrt{S_K}})e^{-\frac{m\epsilon^2}{2M^2}} \tag{19}$$

which is of the form $Pr(X \ge 2\epsilon) \le a$. We take the probability of its complement. From eq.(19), therefore, we can obtain $Pr(X \le 2\epsilon) \ge 1 - a$, which would be $Pr(\sup_{f \in \mathcal{H}}|\ell_z(f)| \le 2\epsilon) \ge 1 - 2\mathcal{N}(\mathcal{H}, \frac{\epsilon}{2\sqrt{S_K}})e^{-\frac{m\epsilon^2}{2M^2}}$ Then, we finally set $\epsilon = \epsilon/2$ to obtain,

$$\text{Prob.}(\sup_{f \in \mathcal{H}}|\ell_z(f)| \le \epsilon) \ge 1 - 2\mathcal{N}(\mathcal{H}, \frac{\epsilon}{4\sqrt{S_K}})e^{-\frac{m\epsilon^2}{4M^2}}$$

which proves the lemma.

On M-boundedness of $f_{\mathcal{H}_K}^m$

To prove the lemma, we assumed that $f_{\mathcal{H}_K}^m$ is M bounded. To see why this is reasonable, from eq.15, we have $||f_{\mathcal{H}_K}^m||_\infty \leq \sqrt{S_K}||f_{\mathcal{H}_K}^m||_{\mathcal{H}_K} \leq \sqrt{S_K}||g||_{\mathcal{L}_2(d\rho)}$. Therefore, $f_{\mathcal{H}_K}^m$ is bounded if $S_K$ and $||g||_{\mathcal{L}_2(d\rho)}$ are bounded, which is true by Proposition 1. □

**Remark 1.** *We derived the error bound based on the Hoeffding's inequality by assuming that our only knowledge about $f$ is that it is bounded. If we have other knowledge, for example, if we know the variance of $f$, we could use Bernstein's inequality instead of Hoeffding's inequality with minimal change to the proof. To the extent we are interested in the contribution of neural network in error bound, however, there is not much gain by using one inequality or the other. Hence, we stick with Hoeffding's inequality and note other possibilities.*

**Remark 2.** *Note that in Lemma 1, the radius of disks are inversely related to the the quantity, $S_K$, meaning that if $S_K$ is high, we would need large number of disks to fill the RKHS space. Hence, it denotes a quantity that reflects the complexity of the RKHS space. We, therefore, term it kernel complexity. Also in eq. 15 and the discussion about the M-boundedness, we see that the maximum value $|f(x)|$ depends on $S_K$, again providing insight into how $S_K$ may control both maximum fluctuation and the boundedness.*

Lemma 1 bounds the probability of error in terms of the covering number of the RKHS space. Next, we use Lemma 2 due to [1] to obtain an error bound in estimating KL divergence with finite samples in Theorem 2.

**Lemma 2** ([1]). *Let $K : \mathcal{X} \times \mathcal{X} \to \mathbb{R}$ be a $\mathcal{C}^\infty$ Mercer kernel and the inclusion $I_K : \mathcal{H}_K \hookrightarrow \mathcal{C}(\mathcal{X})$ be the compact embedding defined by $K$ to the Banach space $\mathcal{C}(\mathcal{X})$. Let $B_R$ be the ball of radius $R$ in RKHS $\mathcal{H}_K$. Then $\forall \eta > 0, R > 0, \nu > n$, we have*

$$\ln \mathcal{N}(I_K(B_R), \eta) \leq \left(\frac{RC_\nu}{\eta}\right)^{\frac{2n}{\nu}} \tag{20}$$

*where $\mathcal{N}$ gives the covering number of the space $I_K(B_R)$ with discs of radius $\eta$, and $n$ represents the dimension of the input space $\mathcal{X}$. $C_\nu$ is given by $C_\nu = C_s\sqrt{||\mathscr{L}_s||}$ where $\mathscr{L}_s$ is a linear embedding from square integrable space $\mathcal{L}_2(d\rho)$ to the Sobolev space $H^{\nu/2}$, $||\mathscr{L}_s||$ denotes operator norm and $C_s$ is a constant.*

To prove Lemma 2, the RKHS space is embedded in the Sobolev Space $H^{\nu/2}$ using $\mathscr{L}_K$ and then covering number of Sobolev space is used. Thus the norm of $\mathscr{L}_K$ and the degree of Sobolev space, $\nu/2$, appears in the covering number of a ball in $\mathcal{H}_K$. In Lemma 3, we use this Lemma to bound the estimation error of KL divergence.

**Lemma 3.** *Let $KL(f_{\mathcal{H}_{K_\theta}}^m)$ and $KL_m(f_{\mathcal{H}_{K_\theta}}^m)$ be the estimates of KL divergence obtained by using true distribution $p(x)$ and $m$ samples respectively and using a fixed kernel, $K_\theta$ as described in Lemma 1, then the probability of error in the estimation at the error level $\epsilon$ is given by:*

$$Prob.(|KL_m(f_{\mathcal{H}_{K_\theta}}^m) - KL(f_{\mathcal{H}_{K_\theta}}^m)| \geq \epsilon) \leq 2\exp\left[\left(\frac{4RC_s\sqrt{S_K(\theta)||\mathscr{L}_s(\theta)||}}{\epsilon}\right)^{\frac{2n}{\nu}} - \frac{m\epsilon^2}{4M^2}\right]$$

*Proof.* Lemma 2 gives the covering number of a ball of radius $R$ in an RKHS space. In Lemma 1, if we consider the hypothesis space to be a ball of radius $R$, we can apply Lemma 2 in it. Additionally, since we fix the radius of disks to be $\eta = \frac{\epsilon}{4\sqrt{S_K}}$ in Lemma 1, we obtain,

$$\text{Prob.}(|KL_m(f_{\mathcal{H}_{K_\theta}}^m) - KL(f_{\mathcal{H}_{K_\theta}}^m)| \geq \epsilon) \leq 2\exp\left[\left(\frac{4\sqrt{S_{K_\theta}}RC_\nu}{\epsilon}\right)^{\frac{2n}{\nu}} - \frac{m\epsilon^2}{4M^2}\right] \tag{21}$$

Substituting $C_\nu = C_s\sqrt{||\mathscr{L}_{K_\theta}||}$, we obtain,

$$\text{Prob.}(|KL_m(f_{\mathcal{H}_K}^m) - KL(f_{\mathcal{H}_{K_\theta}}^m)| \geq \epsilon) \leq 1 - 2\exp\left[\left(\frac{4RC_s\sqrt{S_{K_\theta}||\mathscr{L}_{K_\theta}||}}{\epsilon}\right)^{\frac{2n}{\nu}} - \frac{m\epsilon^2}{4M^2}\right] \tag{22}$$

□

**Theorem 2.** *Let $KL(f_{\mathcal{H}}^m)$ and $KL_m(f_{\mathcal{H}}^m)$ be the estimates of KL divergence obtained by using true distribution $p(x)$ and $m$ samples respectively as described in Lemma 1, then the probability of error in the estimation at the error level $\epsilon$ is given by:*

$$Prob.(|KL_m(f_{\mathcal{H}}^m) - KL(f_{\mathcal{H}}^m)| \leq \epsilon) \geq 1 - 2\exp\left[\left(\frac{4RC_p\sqrt{S_p||\mathscr{L}_p||}}{\epsilon}\right)^{\frac{2n}{\nu}} - \frac{m\epsilon^2}{4M^2}\right]$$

*where $C_p\sqrt{S_p||\mathscr{L}_p||} = \sup_{K_\theta} C_s\sqrt{S_K(\theta)||\mathscr{L}_s||}$, i.e. $C_p, S_p, \mathscr{L}_p$ correspond to a kernel for which the bound is maximum.*

*Proof.* Lemma 3 gives an error bound for a fixed kernel, $K_\theta$. To find an upper bound over all possible kernels, we take the supremum over all kernels.

$$\text{Prob.}(|KL_m(f_{\mathcal{H}}^m) - KL(f_{\mathcal{H}}^m)| \geq \epsilon) \leq \sup_{K_\theta} \text{Prob.}(|KL_m(f_{\mathcal{H}_{K_\theta}}^m) - KL(f_{\mathcal{H}_{K_\theta}}^m)| \geq \epsilon) \quad (23)$$

$$\leq 2\exp\left[\left(\frac{4RC_p\sqrt{S_p||\mathscr{L}_p||}}{\epsilon}\right)^{\frac{2n}{\nu}} - \frac{m\epsilon^2}{4M^2}\right] \quad (24)$$

where $S_p = S_K(\theta_p)$ and $\mathscr{L}_p = \mathscr{L}_K(\theta_p)$, *i.e.*, $S_p$ and $\mathscr{L}_p$ correspond to kernel complexity and Sobolev operator norm corresponding to optimal kernel $K_{\theta_p}$ that extremizes eq. (23). Theorem statement readily follows from eq. (24) $\qquad\square$

# 7 Variance and Consistency of the Estimate

## 7.1 Variance Analysis

**Theorem 3.** *Let $X = KL_m(f_{\mathcal{H}}^m)$ be the estimated KL divergence using m samples as described in Theorem 2. Assuming that $X$ follows a Gaussian distribution $X \sim \mathcal{N}(\mu, \varsigma^2)$, we can obtain an upper bound on this variance of the estimate as follows:*

$$\varsigma \leq \frac{\epsilon}{\sqrt{2}erf^{-1}\left[-4\exp\left[\left(\frac{4RC_p\sqrt{S_p||\mathscr{L}_p||}}{\epsilon}\right)^{\frac{2n}{\nu}} - \frac{m\epsilon^2}{4M^2}\right] + 1\right]} \quad (25)$$

*where erf is the Gauss error function*

$$erf(x) = \frac{2}{\sqrt{\pi}}\int_0^x e^{-t^2}dt \quad (26)$$

*and it is a monotonic function.*

*Proof.* $X$ follows a Gaussian distribution with mean $\mu$ and variance $\varsigma^2$. Let its cumulative distribution function be $\Phi_{\mu,\varsigma}$. By definition,

$$P(X \leq \hat{x}) = \Phi_{\mu,\varsigma}(\hat{x}) \quad (27)$$
$$P(X \geq \hat{x}) = 1 - \Phi_{\mu,\varsigma}(\hat{x}) \quad (28)$$
$$P(X - \mu \geq \epsilon) = 1 - \Phi_{\mu,\varsigma}(\mu + \epsilon) \quad (29)$$

Since two sided probability is higher than one sided, we have,

$$P(X - \mu \geq \epsilon) \leq P(|X - \mu| \geq \epsilon) \quad (30)$$

$$\leq 2\exp\left[\left(\frac{4RC_s\sqrt{S_K||\mathscr{L}_s||}}{\epsilon}\right)^{\frac{2n}{\nu}} - \frac{m\epsilon^2}{4M^2}\right] \quad (31)$$

where we used Theorem 2. Using eq.29, we have,

$$1 - \Phi_{\mu,\varsigma}(\mu + \epsilon) \leq 2\exp\left[\left(\frac{4RC_s\sqrt{S_K||\mathscr{L}_s||}}{\epsilon}\right)^{\frac{2n}{\nu}} - \frac{m\epsilon^2}{4M^2}\right] \quad (32)$$

For a Gaussian distribution, we can use the following expression for the cumulative distribution function,

$$\Phi_{\mu,\varsigma}(\hat{x}) = \frac{1}{2}\Big[1 + \text{erf}\big(\frac{\hat{x} - \mu}{\varsigma\sqrt{2}}\big)\Big] \tag{33}$$

where erf is the Gauss error function. Using this in the eq.32,

$$1 - \text{erf}\big(\frac{\epsilon}{\varsigma\sqrt{2}}\big) \leq 4\exp\left[\left(\frac{4RC_s\sqrt{S_K\|\mathscr{L}_s\|}}{\epsilon}\right)^{\frac{2n}{\nu}} - \frac{m\epsilon^2}{4M^2}\right] \tag{34}$$

$$\text{erf}\big(\frac{\epsilon}{\varsigma\sqrt{2}}\big) \geq -4\exp\left[\left(\frac{4RC_s\sqrt{S_K\|\mathscr{L}_s\|}}{\epsilon}\right)^{\frac{2n}{\nu}} - \frac{m\epsilon^2}{4M^2}\right] + 1 \tag{35}$$

Since the function erf is invertible within domain (-1,1), we have,

$$\frac{\epsilon}{\varsigma\sqrt{2}} \geq \text{erf}^{-1}\left[-4\exp\left[\left(\frac{4RC_s\sqrt{S_K\|\mathscr{L}_s\|}}{\epsilon}\right)^{\frac{2n}{\nu}} - \frac{m\epsilon^2}{4M^2}\right] + 1\right] \tag{36}$$

$$\varsigma \leq \frac{\epsilon}{\sqrt{2}\text{erf}^{-1}\left[-4\exp\left[\left(\frac{4RC_s\sqrt{S_K\|\mathscr{L}_s\|}}{\epsilon}\right)^{\frac{2n}{\nu}} - \frac{m\epsilon^2}{4M^2}\right] + 1\right]} \tag{37}$$

□

Note that Theorem 3 makes a strong assumption that the estimate is distributed as a Gaussian distribution. While it gives us good intuition about the decay of variance as the complexity increases, it is natural to inquire about the validity of this type of relation in a more general sense without assumption on the probability distribution of the estimate. To make a general statement, the key idea is to understand how the cumulative distribution function (CDF) is related to the variance. To clarify this point further, let's look at the eq.(33): $1 - \Phi_{\mu,\varsigma}(\mu + \epsilon) \leq 2\exp\left[\left(\frac{4RC_s\sqrt{S_K\|\mathscr{L}_s\|}}{\epsilon}\right)^{\frac{2n}{\nu}} - \frac{m\epsilon^2}{4M^2}\right]$.

This equation connects the CDF to the variables like $S_K$, $R$, $\mathscr{L}_s$ of the discriminator function space without assuming anything about the shape of the distribution. For a Gaussian distribution, we plug in CDF of a Gaussian distribution, $\Phi_{\mu,\varsigma}(\hat{x}) = \frac{1}{2}\Big[1 + \text{erf}\big(\frac{\hat{x} - \mu}{\varsigma\sqrt{2}}\big)\Big]$ and obtain the result of Theorem 3. For any other distribution, we can carry out similar analysis. A key factor that determines the behavior between the variance and the discriminator complexity is how variance appears in the CDF expression. For example, in both Gaussian distribution and in exponential distribution, we know that the relation between CDF function and variance is inversely related. We believe that as long as this inverse type of relation between CDF and variance holds, we can obtain statements like Theorem 3 for other distributions as well. This leads to a key insight: *similar to the Gaussian case, the decaying behavior of the variance with decreasing complexity holds as long as the inverse relation between the CDF function and the variance holds.*

### 7.2 Consistency of Estimates

**Theorem 4.** *Let $f^*$ and $f_h^m$ and $f_h^*$ be optimal discriminators defined as*

$$f^* = \underset{f}{\text{argmax}}[E_{p(x)}\log\sigma(f(x)) + E_{q(x)}\log(1 - \sigma(f(x)))] \tag{38}$$

$$f_h^* = \underset{f\in h}{\text{argmax}}[E_{p(x)}\log\sigma(f(x)) + E_{q(x)}\log(1 - \sigma(f(x)))] \tag{39}$$

$$f_h^m = \underset{f\in h}{\text{argmax}}\Big[\frac{1}{m}\sum_{x_i\sim p(x_i)}\log\sigma(f(x_i)) + \frac{1}{m}\sum_{x_j\sim q(x_j)}\log(1 - \sigma(f(x_j)))\Big] - \frac{\lambda_0}{m}\|g\|_{\mathcal{L}_2(d\tau)}^2 \tag{40}$$

*and the KL estimate is given by $KL(f) = E_{p(x)}[f(x)]$, $KL_m(f) = \frac{1}{m}\sum_{x_i\sim p(x_i)}[f(x)]$. Then, in the limiting case as $m \to \infty$, $|KL_m(f_h^m) - KL(f^*)| \xrightarrow{\mathbb{P}} 0$ i.e., the convergence is in probability.*

*Proof.* Estimation error can be divided into three terms as

$$KL_m(f_h^m) - KL(f^*) = \underbrace{KL_m(f_h^m) - KL(f_h^m)}_{Deviation\text{-}from\text{-}mean\ error} + \underbrace{KL(f_h^m) - KL(f_h^*)}_{Discriminator\ induced\ error} + \underbrace{KL(f_h^*) - KL(f^*)}_{Bias}$$

$$(41)$$

Therefore,

$$|KL_m(f_h^m) - KL(f^*)| \leq |KL_m(f_h^m) - KL(f_h^m)| + |KL(f_h^m) - KL(f_h^*)|$$
$$+ |KL(f_h^*) - KL(f^*)| \quad (42)$$

To show that the total error goes to zero, we show that each term on the right goes to zero. The last term is the bias and we assume that the RKHS space $h = \mathcal{H}$ we consider consists the true solution, $f^*$. Hence the bias goes to zero.

Using Theorem 2, it is immediately clear that the first term, $|KL_m(f_h^m) - KL(f_h^m)|$ approaches zero in probability in the limiting case as $m \to \infty$.

The only remaining is the second term, $|KL(f_h^m) - KL(f_h^*)|$. In Theorem 5 we show that this term also goes to zero almost surely as $m \to \infty$. □

**Theorem 5.** *Let $f_h^*$ and $f_h^*$ be the optimal discriminators as defined in eq. (39) and eq. (40), and the KL divergence estimate using discriminators learned using finite and infinite samples be $KL(f_h^m) = \int [f_h^m(x)]p(x)dx$ and $KL(f_h^*) = \int [f_h^*(x)]p(x)dx$, where, Then, in the limiting case, we have*

$$\lim_{m \to \infty} |KL(f_h^m) - KL(f_h^*)| = 0$$

*Proof.*

$$|KL(f_h^m) - KL(f_h^*)| = |\int [f_h^m(x) - f_h^*(x)]p(x)dx|$$
$$\leq \sup_x |f_h^m(x) - f_h^*(x)| = ||f_h^m(x) - f_h^*(x)||_\infty$$

Therefore, we can show $\lim_{m \to \infty} KL(f_h^m) - KL(f_h^*) = 0$ if $\lim_{m \to \infty} ||f_h^m(x) - f_h^*(x)||_\infty = 0$, that is, if the function $f_h^m(x)$ converges uniformly to function $f_h^*(x)$ in the limiting case.

The two maximizer functions are given by

$$f_h^* = \underset{f \in h}{\mathrm{argmax}}[E_{p(x)} \log \sigma(f(x)) + E_{q(x)} \log(1 - \sigma(f(x)))] \quad (43)$$

$$f_h^m = \underset{f \in h}{\mathrm{argmax}}\Big[\frac{1}{m}\sum_{x_i \sim p(x_i)} \log \sigma(f(x_i)) + \frac{1}{m}\sum_{x_j \sim q(x_j)} \log(1 - \sigma(f(x_j)))\Big] - \frac{\lambda_0}{m}||g||^2 \quad (44)$$

As a first step in showing that $f_h^m$ uniformly approaches $f_h^*$, we first show that $\lim_{m \to \infty} \frac{\lambda_0}{m}||g||^2 = 0$ in Lemma 4.

Then, to prove the rest, let us denote,

$$G_m(f) = \frac{1}{m}\sum_{x_i \sim p(x_i)} \log \sigma(f(x_i)) + \frac{1}{m}\sum_{x_j \sim q(x_j)} \log(1 - \sigma(f(x_j)))$$

$$G(f) = E_{p(x)} \log \sigma(f(x)) + E_{q(x)} \log(1 - \sigma(f(x)))$$

In Lemma 5, we prove that functionals $G(f)$ and $G_m(f)$ are concave with respect to function $f$. In the light of these two lemmas, we argue

$$\lim_{m \to \infty} ||f_h^m(x) - f_h^*(x)||_\infty = 0 \text{ if } \lim_{m \to \infty} \sup_f |G_m(f) - G(f)| = 0 \quad (45)$$

Next, we show $\lim\limits_{m\to\infty}\sup\limits_f |G_m(f) - G(f)| = 0$ as follows. We have,

$$
\begin{aligned}
|G_m - G| = &|\frac{1}{m}\sum_{x_i\sim p(x_i)}\log\sigma(f(x_i)) + \frac{1}{m}\sum_{x_j\sim q(x_j)}\log(1-\sigma(f(x_j))) \\
&- E_{p(x)}\log\sigma(f(x)) + E_{q(x)}\log(1-\sigma(f(x)))|
\end{aligned}
\tag{46}
$$

$$
\begin{aligned}
\leq &|\frac{1}{m}\sum_{x_i\sim p(x_i)}\log\sigma(f(x_i)) - E_{p(x)}\log\sigma(f(x))| \\
&+ |\frac{1}{m}\sum_{x_j\sim q(x_j)}\log(1-\sigma(f(x_j))) - E_{q(x)}\log(1-\sigma(f(x)))|
\end{aligned}
\tag{47}
$$

$$
\begin{aligned}
\lim_{m\to\infty}\sup_f |G_m(f) - G(f)| \leq &\lim_{m\to\infty}\sup_f |\frac{1}{m}\sum_{x_i\sim p(x_i)}\log\sigma(f(x_i)) - E_{p(x)}\log\sigma(f(x))| \\
&+ \lim_{m\to\infty}\sup_f |\frac{1}{m}\sum_{x_j\sim q(x_j)}\log(1-\sigma(f(x_j))) - E_{q(x)}\log(1-\sigma(f(x)))|
\end{aligned}
\tag{48}
$$

Both the terms on right hand side go to zero if $\log\circ\sigma\circ f$ is in a Glivenko Cantelli class of functions using Empirical Process Theory [10], which we prove in Lemma 6. That completes the proof. $\quad\square$

**Lemma 4.** $\lim\limits_{m\to\infty}\frac{\lambda_0}{m}||g||^2 = 0$

*Proof.* $||g||_{\mathcal{L}_2(d\rho)}$ is bounded because $g$ is Lipschitz continuous and its domain is bounded. Since, $||g||_{\mathcal{L}_2(d\rho)}$ is bounded, we immediately obtain the required statement. $\quad\square$

**Lemma 5.** *The functional $G(f)$ is concave with respect to function $f$ in the following sense: $\theta_1 G(f_1) + \theta_2 G(f_2) \leq G(\theta_1 f_1 + \theta_2 f_2)$ for any $\theta_1, \theta_2 \in (0,1)$ such that $\theta_1 + \theta_2 = 1$. The same is true for $G_m(f)$.*

*Proof.*

$$
\begin{aligned}
\theta_1 G(f_1) + \theta_2 G(f_2) = &\theta_1\Big[\int p(x)\log\sigma(f_1(x))dx + \int q(x)\log(1-\sigma(f_1(x)))dx\Big] \\
&+ \theta_2\Big[\int p(x)\log\sigma(f_2(x))dx + \int q(x)\log(1-\sigma(f_2(x)))dx\Big]
\end{aligned}
\tag{49}
$$

$$
\begin{aligned}
= &\int p(x)\Big[\theta_1\log\sigma(f_1(x))dx + \theta_2\log\sigma(f_2(x))dx\Big] \\
&+ \int q(x)\Big[\theta_1\log\sigma(-f_1(x))dx + \theta_2\log\sigma(-f_2(x))dx\Big]
\end{aligned}
\tag{50}
$$

$$
\begin{aligned}
\leq &\int p(x)\log\sigma[\theta_1 f_1(x) + \theta_2 f_2(x)]dx \\
&+ \int q(x)\log\sigma[-(\theta_1 f_1(x) + \theta_2 f_2(x))]dx
\end{aligned}
\tag{51}
$$

$$
= G(\theta_1 f_1 + \theta_2 f_2)
\tag{52}
$$

where we used the fact that $\log(1-\sigma(f(x))) = \log\sigma(-f(x))$ (this is straightforward using definition of Sigmoid function, $\sigma$) in line 50. In line 51, we used the fact that $\log\sigma$ is a concave function (see Lemma 8). $\quad\square$

**Lemma 6.** $\log\circ\sigma\circ f$ *is a Glivenko Cantelli class of function.*

*Proof.* In Lemma 7, we show that, by definition, $f$ is Lipschitz continuous with some Lipschitz constant $L_f$. In Lemma 8 we show that if $f$ is a Lipschitz continuous function from $\mathcal{X}$ to $(-\infty,\infty)$ with Lipschitz constant, $L_f$, then $\log\sigma f$ is a function from $\mathcal{X}$ to $(-\infty,0)$ with same Lipschitz constant $L_f$. Hence, $v = \log\sigma f$ is a a function from $\mathcal{X}$ to $(-r,0)$. Note that since $\mathcal{X}$ is bounded

and $f$ is Lipschitz continuous from $\mathcal{X}$ to $\mathbb{R}$, we can always find some $r$ such that $v$ maps from $\mathcal{X}$ to $(-r, 0)$.

Now, we show that $v = \log \sigma f$ is Glivenko Cantelli by entropy number. Let $\mathcal{V} = \{v : v = \log(\sigma(f)), f \in \mathcal{F}\}$. In Lemma 10, we use theorem from [10] to show that $\mathcal{V}$ is Glivenko Cantelli if and only if

$$\frac{1}{m} \log N(\epsilon, \mathcal{V}_M, \ell_1(\mathbb{P}_m)) \xrightarrow{\mathbb{P}} 0, \tag{53}$$

for any $M > 0, \epsilon$, where $\mathcal{V}_M$ is the class of functions $v\mathbf{1}\{E \leq M\}$ where $v$ ranges over $\mathcal{V}$ and $E$ is an envelope function to $\mathcal{V}$. Since we proved that $\log(\sigma(f)(x)) < 0$ for any $x$, we can choose $E = v_0(x) = \mathbf{0}$ as a constant function that is an envelope to $\mathcal{V}$. For any $M > 0$, therefore, $1\{E \leq M\} = 1$ trivially and $\mathcal{V}_M = \mathcal{V}$. Hence, we just need to show

$$\frac{1}{m} \log N(\epsilon, \mathcal{V}, \ell_1(\mathbb{P}_m)) \xrightarrow{\mathbb{P}} 0 \tag{54}$$

In Lemma 9, we show that the entropy number of such a function is given by

$$\log \mathcal{N}(\epsilon, \mathcal{V}, \ell_1(\mathbb{P}_m)) \leq \left( \frac{16 L.diam(\mathcal{X})}{\epsilon} \right)^{ddim(\mathcal{X})} \log \left( \frac{4r}{\epsilon} \right) \tag{55}$$

and therefore is bounded and independent of the sample size $m$. Hence, $\frac{1}{m} \log N(\epsilon, \mathcal{V}, \ell_1(\mathbb{P}_m))$ goes to 0. $\qquad\square$

**Lemma 7.** *The function $f$ defined in Theorem 1 on the main paper as:*

$$f(x) = \int_{\mathcal{W}} g(w)\psi(x, w)d\tau(w), \tag{56}$$

*where $\psi(x, w) = \phi_\theta(x)^T w$ and the function $\phi_\theta$ is Lipschitz continuous with Lipschitz constant $L_\phi$. Then, the function $f$ is Lipschitz continuous with some Lipschitz constant, $L_f$.*

*Proof.* By the definition,

$$f(x) = \langle g(w), \psi(x, w) \rangle_{\mathcal{L}_2(d\tau)} \tag{57}$$

For any two points $x_1$ and $x_2$,

$$|f(x_1) - f(x_2)| = \langle g(w), \psi(x_1, w) - \psi(x_2, w) \rangle_{\mathcal{L}_2(d\tau)} \tag{58}$$

$$\leq ||g(w)||_{\mathcal{L}_2(d\tau)} ||\psi(x_1, w) - \psi(x_2, w)||_{\mathcal{L}_2(d\tau)} \tag{59}$$

where we used Cauchy Schwartz. Now, taking the difference in $\psi$, it can be written as

$$||\psi(x_1, w) - \psi(x_2, w)||_{\mathcal{L}_2(d\tau)} = \sqrt{\int [\psi(x_1, w) - \psi(x_2, w)]^2 d\tau(w)} \tag{60}$$

$$= \sqrt{\int [(\phi_\theta(x_1) - \phi_\theta(x_2))^T w]^2 d\tau(w)} \tag{61}$$

$$\leq \sqrt{\int ||\phi_\theta(x_1) - \phi_\theta(x_2)||^2 ||w||^2 d\tau(w)} \tag{62}$$

where we again used Cauchy Schwartz in the last line since $[(\phi_\theta(x_1) - \phi_\theta(x_2))^T w]$ is an inner product in $\mathbb{R}^D$ where D is the dimension of $w$. Since $\phi_\theta$ is Lipschitz continuous with Lipschitz constant $L_\phi$, we have

$$||\phi_\theta(x_1) - \phi_\theta(x_2)|| \leq L_\phi ||x_1 - x_2||$$

Using this inequality in eq.62, we obtain

$$||\psi(x_1, w) - \psi(x_2, w)||_{\mathcal{L}_2(d\tau)} \leq L_\phi ||x_1 - x_2|| \sqrt{\int ||w||^2 d\tau(w)} \tag{63}$$

$$= L_\phi ||x_1 - x_2|| \sqrt{tr(C_w)} \tag{64}$$

where, $C_w$ is the uncentered covariance matrix of Gaussian distributed $w$. Plugging eq.(64) in eq.(59), we obtain

$$|f(x_1) - f(x_2)| \leq ||g(w)||_{\mathcal{L}_2(d\tau)} L_\phi \sqrt{tr(C_w)} ||x_1 - x_2|| \tag{65}$$

Since, we have that $||g(w)||_{\mathcal{L}_2(d\tau)} < \infty$ (see Lemma 4), we have proved that $f$ is Lipschitz continuous with Lipschitz constant given by $L_f \leq ||g(w)||_{\mathcal{L}_2(d\tau)} L_\phi \sqrt{tr(C_w)}$. $\qquad \square$

**Lemma 8.** *The function* $\log \circ \sigma$ *exhibits following properties:*
*i) It is a concave function with its derivative always between* $0$ *and* $1$
*ii) If the Lipschitz constant of* $f$ *is* $L_f$*, so is the Lipschitz constant of* $\log \circ \sigma \circ f$

*Proof.* i) Let us denote $u(x) = \log(\sigma(x))$. Then, we have,

$$u(x) = \log \frac{e^x}{1 + e^x} = x - \log(1 + e^x) \tag{66}$$

$$u'(x) = 1 - \frac{e^x}{1 + e^x} = \frac{1}{1 + e^x} \tag{67}$$

$$0 < u'(x) < 1, \quad \forall x \in (-\infty, \infty) \tag{68}$$

which proves that the derivative is between $0$ and $1$. To show that $u(x)$ is concave, it is sufficient to note that its second derivative is always negative.

ii) Let us use notation $u = \log(\sigma)$, and let $f_2 = f(x_2)$, $f_1 = f(x_1)$, $u_2 = u(f(x_2))$, $u_1 = u(f(x_1))$. Since the maximum derivative of $u$ is upper bounded by 1, $u$ as a function of $f$ has Lipschitz constant 1 and therefore, we can write

$$u_2 - u_1 = u(f_2) - u(f_1) \leq f_2 - f_1 = f(x_2) - f(x_1) \tag{69}$$

$$\leq L_f ||x_2 - x_1|| \tag{70}$$

where the last inequality is because $f$ is Lipschitz continuous with Lipschitz constant $L_f$. This proves that the Lipschitz constant of $\log \circ \sigma \circ f$ is also $L_f$.

$\qquad \square$

**Lemma 9.** *Let* $\mathcal{F}_L$ *be the space of L-Lipschitz functions mapping the metric space* $(\mathcal{X}, \rho)$ *to [0,r].*
*Let* $ddim(\mathcal{X})$ *and* $diam(\mathcal{X})$ *denote the doubling dimension and diameter of* $\mathcal{X}$ *respectively. Then,*
*i) the covering numbers of* $\mathcal{F}_L$ *can be estimated in terms of the covering numbers of* $\mathcal{X}$*:*

$$\mathcal{N}(\epsilon, \mathcal{F}_L, ||.||_\infty) \leq \left(\frac{4r}{\epsilon}\right)^{\mathcal{N}(\epsilon/8L, \mathcal{X}, ||.||_\infty)} \tag{71}$$

*ii) the entropy number of* $\mathcal{F}_L$ *can be estimated as:*

$$\log \mathcal{N}(\epsilon, \mathcal{F}_L, ||.||_\infty) \leq \left(\frac{16L.diam(\mathcal{X})}{\epsilon}\right)^{ddim(\mathcal{X})} \log\left(\frac{4r}{\epsilon}\right) \tag{72}$$

*iii) the entropy number with respect to* $\ell_1(\mathbb{P}_m) = \int |f| d\mathbb{P}_m = \frac{1}{m} \sum_k |f(x_k)|$ *defined with respect to the* $m$ *input points, is the same as (ii), i.e.*

$$\log \mathcal{N}(\epsilon, \mathcal{F}_L, \ell_1(\mathbb{P}_m)) \leq \left(\frac{16L.diam(\mathcal{X})}{\epsilon}\right)^{ddim(\mathcal{X})} \log\left(\frac{4r}{\epsilon}\right) \tag{73}$$

*where* $\mathbb{P}_m$ *is an empirical probability measure with respect to* $m$ *inputs points in* $\mathcal{X}$*.*

*Proof.* The proof is adapted from [4] Lemma 2 and [3] Lemma 6, and modified to handle range $[0, r]$.

i) We first cover the domain $\mathcal{X}$ by $N$ balls $U_1, U_2, ..., U_{|N|}$, where $N = \mathcal{N}(\epsilon/8L, \mathcal{X}, ||.||_\infty)$ is the covering number of $\mathcal{X}$, $N = \{x_i \in U_i\}_{i=1}^{|N|}$ is a set of center points of $|N|$ balls and $\epsilon' = \epsilon/8L$ is the radius of the covering balls.

Now, our strategy is to construct an $\epsilon$ cover $\hat{F} = \{\hat{f}_1, ..., \hat{f}_{|\hat{F}|}\}$ for $\mathcal{F}_L$ with respect to $||.||_\infty$. To do so, at every point $x_i \in N$, we choose the value of $\hat{f}(x_i)$ to be some multiple of $2L\epsilon' = \frac{\epsilon}{4}$, while

maintaining $||\hat{f}||_{Lip} \leq 2L$. We then construct a 2L-Lipschitz extension for $\hat{f}$ from $N$ to all over $\mathcal{X}$(note that such an extension always exists, see [6, 11]).

With this construction, we can show that every $f \in \mathcal{F}_L$ is close to some $\hat{f} \in \hat{F}$ in the sense that $||f - \hat{f}||_{\infty} \leq \epsilon$. To show this, note the following:

$$|f(x) - \hat{f}(x)| \leq |f(x) - f(x_N)| + |f(x_N) - \hat{f}(x_N)| + |\hat{f}(x_N) - \hat{f}(x)| \tag{74}$$

$$\leq L.\rho(x, x_N) + \epsilon/4 + 2L.\rho(x, x_N) \tag{75}$$

$$\leq \epsilon \tag{76}$$

where the inequality in eq.75 is due to the fact that $f$ is $L$-Lipschitz and $\hat{f}$ is $2L$-Lipschitz and since we have covered the input space $\mathcal{X}$, each $x$ is within $\epsilon'$ of some $x_N$. Also note that for every $f(x_N)$ we can find $\hat{f}(x_N)$ within some radius $\epsilon/4$; this is because we choose $f(x_N)$ to be some multiple of $2L\epsilon'$. Finally, we need to compute the cardinality of $\hat{F}$, i.e. $|\hat{F}|$. For any $x_i \in |N|$, $\hat{f}$ can take one of the multiple of $2L\epsilon'$ values. Hence, there are $r/2L\epsilon'$ such possibilities as the range is $[0, r]$. Since there are $|N|$ such possibilities for $x_i$, the upper bound on all possible function values $\hat{f}$ is $(\frac{r}{2L\epsilon'})^{|N|} = (\frac{4r}{\epsilon})^{|N|}$, which proves the first statement after plugging in the value of $|N|$.

ii) Taking logarithm of the result in i)

$$\log \mathcal{N}(\epsilon, \mathcal{F}_L, ||.||_{\infty}) \leq \mathcal{N}(\epsilon/8L, \mathcal{X}, ||.||_{\infty}) \log\left(\frac{4r}{\epsilon}\right) \tag{77}$$

The covering number of the input space, $\mathcal{X}$ in terms of doubling dimension, $ddim(\mathcal{X})$ and diameter, $diam(\mathcal{X})$ can be written as [5]:

$$\mathcal{N}(\epsilon, \mathcal{X}, ||.||_{\infty}) \leq \left(\frac{2diam(\mathcal{X})}{\epsilon}\right)^{ddim(\mathcal{X})} \tag{78}$$

Plugging this expression in eq.(77), we obtain the required expression.

iii) The result in i) is with respect to $||.||_{\infty}$. In eq.(75), we showed that for any $f \in \mathcal{F}_L$ there is some $\hat{f} \in \hat{F}$ within a radius of $\epsilon$ such that $||f - \hat{f}||_{\infty} \leq \epsilon$. Here, we show that this also implies that $||f - \hat{f}||_{\ell_1(\mathbb{P}_m)} \leq \epsilon$. We show this as follows:

$$||f - \hat{f}||_{\ell_1(\mathbb{P}_m)} = \frac{1}{m} \sum_{k=1}^{m} |f(x_k) - \hat{f}(x_k)| \tag{79}$$

$$\leq \frac{1}{m} \sum_{k=1}^{m} \epsilon = \epsilon \tag{80}$$

Therefore, the entropy number with respect to $\ell_1(\mathbb{P}_m)$ metric is same as the entropy number with respect to the $||.||_{\infty}$, which proves our third claim. □

**Lemma 10** ([10] Theorem 3.5. ). *Let $\mathcal{V}$ be a class of measurable functions with envelope $E$ such that $P(E) < \infty$. Let $\mathcal{V}_M$ be the class of functions $v.\mathbf{1}\{E \leq M\}$ where $v$ ranges over $\mathcal{V}$. Then, $\mathcal{V}$ is a Glivenco Cantelli class of functions, i.e. it satisfies*

$$\sup_{v \in \mathcal{V}} |\mathbb{P}_m v - Pv| \tag{81}$$

*, if and only if*

$$\frac{1}{m} \log N(\epsilon, \mathcal{V}_M, L_1(\mathbb{P}_m)) \xrightarrow{\mathbb{P}} 0, \tag{82}$$

*for every $\epsilon > 0$ and $M > 0$, where $Pv = \int vdP$ and $\mathbb{P}_m v = \frac{1}{m} \sum_k v(x_k)$.*

# 8    Experimental Results

**Code:**    The code is made publicly available at https://github.com/sandeshgh/Reliable-KL-estimation Below we describe some aspects of our implementation.

## 8.1 Two Gaussian

### 8.1.1 Architecture and Implementation

RKHS Discriminator Architecture (Pytorch Code)

```python
class RKHS_Net(nn.Module):
    def __init__(self, dim =10, mid_dim1=20, mid_dim2=20, mid_dim3=20, D=50, gamma
        =1, metric = 'rbf', lip=5, g_lip =5):
        super(RKHS_Net, self).__init__()
        self.gamma = torch.FloatTensor([gamma])
        self.metric = metric
        self.D = D
        self.act = nn.ReLU()
        self.lin1 = spectral_norm( nn.Linear(dim, mid_dim1), k =g_lip)
        self.lin2 = spectral_norm( nn.Linear(mid_dim1, mid_dim2), k =g_lip)
        self.lin3 = spectral_norm( nn.Linear(mid_dim2 , mid_dim3), k =g_lip)
        self.lin4 = spectral_norm( nn.Linear(mid_dim3, 1), k =g_lip)

        self.g = nn.Sequential(self.lin1,
                               self.act,
                               self.lin2,
                               self.act,
                               self.lin3,
                               self.act,
                               self.lin4
                               )

        self.lin_phi1 = spectral_norm(nn.Linear(2, mid_dim1), k=lip)
        self.lin_phi2 = spectral_norm(nn.Linear(mid_dim1, mid_dim2), k=lip)
        self.lin_phi3 = spectral_norm(nn.Linear(mid_dim2, mid_dim3), k=lip)
        self.lin_phi4 = spectral_norm(nn.Linear(mid_dim3, dim), k=lip)

        self.phi = nn.Sequential(self.lin_phi1,
                                 self.act,
                                 self.lin_phi2,
                                 self.act,
                                 self.lin_phi3,
                                 self.act,
                                 self.lin_phi4
                                 )

    def forward(self, y):
        x=self.phi(y)
        d = x.shape[1]
        if self.metric =='rbf':
            w= torch.sqrt(2*self.gamma)*torch.randn(size=(self.D,d))
        w=w.to(x.device)
        psi = ((torch.matmul(x,w.permute(1,0))
            ))*(torch.sqrt(2/torch.FloatTensor([self.D])).to(x.device))
        w_a = w
        g= self.g(w_a)
        f = (psi*g.permute(1,0)).mean(1)
        g_norm =(g**2).mean()
        return f, g_norm
```

Simple Neural Network Discriminator Architecture ( Pytorch Code)

```python
class DNet_basic(nn.Module):
    def __init__(self, input_dim, mid_dim1, mid_dim2, output_dim, lip_constraint =
        False, lip = 5):
        super(DNet_basic, self).__init__()

        self.act = nn.ReLU()
```

```
        self.lin1 = nn.Linear(input_dim, mid_dim1)
        self.lin2 = nn.Linear(mid_dim1, mid_dim2)
        self.lin3 = nn.Linear(mid_dim2, mid_dim2)
        self.lin4 = nn.Linear(mid_dim2, output_dim)
        # self.sigmoid=nn.Sigmoid()

        self.phi = nn.Sequential(self.lin1,
                                 self.act,
                                 self.lin2,
                                 self.act,
                                 self.lin3,
                                 self.act,
                                 self.lin4
                                 )

    def forward(self, x):
        t = self.phi(x)
        return t
```

**Discrete approximation:** Both the discriminators have stacked Fully connected layers and activation function. In the proposed RKHS discriminator, we have an additional network `self.g` which we use to approximate the continuous integral $f(x) = \int_{\mathcal{W}} g(w)\psi(x,w)d\tau(w)$ with the following discrete approximation:

$$f(x) = \frac{1}{D}\sum_{k=1}^{D} g(w_k)(\phi^T w_k \sqrt{\frac{2}{D}}) \tag{83}$$

where $w$ is sampled from a Normal distribution with variance $\gamma$. In our experiments $D = 500$ was sufficient. Note that the Neural network discriminator is similar to $\phi^T w$, except that $w$ is not randomly sampled and there is no $g$ network.

**Lipschitz constraints:** : To enforce Lipschitz constraints on network $g$ and $\phi$ consistent with our assumptions and theoretical results, we use spectral normalization in the RKHS discriminator while it is absent in the basic Neural network discriminator.

### 8.1.2 Data and Hyperparameters

**Data**: Since this is a toy experiment, data were generated locally using pytorch command `randn` to sample from Gaussian distribution.

**Learning rate:** $5 \times 10^{-3}$ (both models)
**No. of samples from each distribution:** 2500 (both models)
**Minibatch size:** 50 (both models)
**$\lambda$ :** 0.005 (RKHS disc.)
**Hyperparameter selection:** (RKHS disc.) The hyperparameters like learning rate and $\lambda$ were selected by first estimating KL divergence at a mid value like 13. Then, same value was used in all experiments.

### 8.1.3 Computational Resources and Time

Running one experiment of KL divergence calculation takes 74 s for the basic algorithm while it takes 245 s for the proposed method in a single GeForce GTX 1080 Ti GPU with 11GB memory.

## 8.2 Mutual Information Estimation

### 8.2.1 Models, Architecture and Implementation

RKHS Discriminator Architecture (Pytorch Code)

```
class ConcatLipFeatures(nn.Module):
```

```python
def __init__(self, dim, hidden_dim, layers, activation,lip, gamma =1, metric =
    'rbf', D=500, mid_dim=5, g_lip =2, **extra_kwargs):
    super(ConcatLipFeatures, self).__init__()
    self.gamma = torch.FloatTensor([gamma])
    self.metric = metric
    self.D = D
    self.act = nn.ReLU()
    self.lin1 = spectral_norm( nn.Linear(hidden_dim, mid_dim), k = g_lip)
    self.lin2 = spectral_norm( nn.Linear(mid_dim, mid_dim), k = g_lip)
    self.lin3 = spectral_norm( nn.Linear(mid_dim, mid_dim), k = g_lip)
    self.lin4 = spectral_norm( nn.Linear(mid_dim, 1), k = g_lip)

    self.g = nn.Sequential(self.lin1,
                           self.act,
                           self.lin2,
                           self.act,
                           self.lin3,
                           self.act,
                           self.lin4
                           )
    # output of this layer is d dim features
    self.rkhs_layer = feature_perceptron(dim * 2, hidden_dim, 1, layers,
        activation, lip)

def forward(self, x, y):
    batch_size = x.size(0)
    # Tile all possible combinations of x and y
    x_tiled = torch.stack([x] * batch_size, dim=0)
    y_tiled = torch.stack([y] * batch_size, dim=1)
    # xy is [batch_size * batch_size, x_dim + y_dim]
    xy_pairs = torch.reshape(torch.cat((x_tiled, y_tiled), dim=2), [
                             batch_size * batch_size, -1])
    # Compute features for each x_i, y_j pair.
    phi = self.rkhs_layer(xy_pairs)
    d = phi.shape[1]
    if self.metric == 'rbf':
        w = torch.sqrt(2 * self.gamma) * torch.randn(size=(self.D, d))
    w = w.to(x.device)
    psi = ((torch.matmul(phi, w.permute(1, 0)))) * (torch.sqrt(2 /
        torch.FloatTensor([self.D])).to(x.device))
    w_a = w # torch.cat((w,u.permute(1,0)),1)
    g = self.g(w_a)
    f = (psi * g.permute(1, 0)).mean(1)
    g_norm = (g ** 2).mean()
    return f, g_norm
```

Simple Neural Network Discriminator Architecture (Pytorch Code)

```python
class ConcatCritic(nn.Module):
    def __init__(self, dim, hidden_dim, layers, activation, **extra_kwargs):
        super(ConcatCritic, self).__init__()
        # output is scalar score
        self._f = mlp(dim * 2, hidden_dim, 1, layers, activation)

    def forward(self, x, y):
        batch_size = x.size(0)
        # Tile all possible combinations of x and y
        x_tiled = torch.stack([x] * batch_size, dim=0)
        y_tiled = torch.stack([y] * batch_size, dim=1)
        # xy is [batch_size * batch_size, x_dim + y_dim]
        xy_pairs = torch.reshape(torch.cat((x_tiled, y_tiled), dim=2), [
                                 batch_size * batch_size, -1])
        # Compute scores for each x_i, y_j pair.
        scores = self._f(xy_pairs)
```

```
        return torch.reshape(scores, [batch_size, batch_size]).t()
```

Similar to the previous experiment, the RKHS discriminator and the Neural network discriminator are similar in core design. The main difference lies in that the RKHS discriminator has this inner product construction same as eq.(83) in previous subsection. To achieve this construction, the RKHS discriminator an additional network, `self.g` and enforces Lipschitz constraint through spectral normalization, which are absent in simple Neural network discriminator.

### 8.2.2 Data and Hyperparameters

**Data:** The experimental setup and data generation follow `https://github.com/ermongroup/smile-mi-estimator`.
**Common for all methods**
**batch size:** 64
**no. of layers:** 2
**hidden dim:** 256
**no. of iterations:** 40000
**learning rate:** $5 \times 10^{-4}$

**Specific to the proposed method**
$\gamma$ :5
Lipschitz constant enforced, $L_\phi$ (layer wise): 5
Lipschitz constant enforced, $L_g$ (layer wise): 5

### 8.2.3 Computational Resources and Time

GPU: GeForce RTX 2080 Ti 11 GB

Below, we report time taken by each method to complete an experiment to obtain mutual information between two 20-d Gaussian distributed random variables using $40,000$ samples from each distribution and mutual information increasing stepwise.

Table 1: Time taken to complete one experiment

| CPC | NWJ | SMILE | Ours (RKHS disc.) |
|-----|-----|-------|-------------------|
| 52 s | 48 s | 52 s | 63 s |

### 8.2.4 Existing Assets

We used the code from the repo `https://github.com/ermongroup/smile-mi-estimator` to generate data as well as run baseline mutual information methods. This code corresponds to the Song et al. [9].

## 8.3 Adversarial Variational Bayes

### 8.3.1 Models, Architecture and Implementation

RKHS Discriminator Architecture (Pytorch Code)

```
class Discriminator_RKHS(nn.Module):
    def __init__(self, x_dim, h_dim, z_dim, lip = 5, g_lip = 5, dim = 10, mid_dim1
        = 20, mid_dim2 = 20, mid_dim3 = 20, D=100, gamma =1, metric = 'rbf'):
        super(Discriminator_RKHS, self).__init__()
        self.metric = metric
        self.gamma = torch.FloatTensor([gamma])
        self.D = D
        self.phi = nn.Sequential(
            spectral_norm(nn.Linear(x_dim + z_dim, h_dim), k = lip),
            nn.LeakyReLU(),
            spectral_norm(nn.Linear(h_dim, h_dim), k = lip),
```

```python
            nn.LeakyReLU(),

            spectral_norm(nn.Linear(h_dim, h_dim), k = lip),
            nn.LeakyReLU(),
            spectral_norm(nn.Linear(h_dim, h_dim), k = lip),
            nn.LeakyReLU(),
            spectral_norm(nn.Linear(h_dim, int(h_dim/4)), k = lip)

        )

        self.act = nn.ReLU()
        self.lin1 = spectral_norm(nn.Linear(int(h_dim/4), mid_dim1), k=g_lip)
        self.lin2 = spectral_norm(nn.Linear(mid_dim1, mid_dim2), k=g_lip)
        self.lin3 = spectral_norm(nn.Linear(mid_dim2, mid_dim3), k=g_lip)
        self.lin4 = spectral_norm(nn.Linear(mid_dim3, 1), k=g_lip)

        self.g = nn.Sequential(self.lin1,
                               self.act,
                               self.lin2,
                               self.act,
                               self.lin3,
                               self.act,
                               self.lin4
                               )

    def weight_init(self, mean, std):
        for m in self._modules:
            normal_init(self._modules[m], mean, std)

    def forward(self, y, z):
        y = y.view(y.shape[0], -1)
        y = torch.cat([y, z], 1)
        x =self.phi(y)
        d = x.shape[1]

        if self.metric == 'rbf':
            w = torch.sqrt(2 * self.gamma) * torch.randn(size=(self.D, d))
        w = w.to(x.device)
        psi = ((torch.matmul(x, w.permute(1, 0)))) * (torch.sqrt(2 /
            torch.FloatTensor([self.D])).to(x.device))
        w_a = w
        g = self.g(w_a)
        f = (psi * g.permute(1, 0)).mean(1)
        g_norm = (g ** 2).mean()
        return f, g_norm
```

Simple Neural Network Discriminator Architecture (Pytorch Code)

```python
class Discriminator_simple(nn.Module):
    def __init__(self, x_dim, h_dim, z_dim):
        super(Discriminator_simple, self).__init__()
        self.net = nn.Sequential(
            nn.Linear(x_dim + z_dim, h_dim),
            nn.LeakyReLU(),
            nn.Linear(h_dim, h_dim),
            nn.LeakyReLU(),

            nn.Linear(h_dim, h_dim),
            nn.LeakyReLU(),
            nn.Linear(h_dim, h_dim),
            nn.LeakyReLU(),
            nn.Linear(h_dim, int(h_dim/4)),
            nn.LeakyReLU(),
            nn.Linear(int(h_dim/4), 1)
```

```
        )

    def weight_init(self, mean, std):
        for m in self._modules:
            normal_init(self._modules[m], mean, std)

    def forward(self, x, z):
        x = x.view(x.shape[0], -1)
        x = torch.cat([x, z], 1)
        out =self.net(x)
        # x = x + torch.sum(z ** 2, 1)
        return out
```

### 8.3.2   Data and Hyperparameters

**Data:** Standard MNIST dataset is used.

**Learning rate:** $10^{-3}$ (both models)
**Minibatch size:** 1024 (both models)
**Hidden dim of encoder/decoder:** 800 (both)
**Hidden dim discriminator:** 1024 (both)
**$\lambda$ :** 1 (RKHS disc.)

### 8.3.3   Computational Resources and Time

GPU: GeForce GTX 1080 Ti 11GB
Time taken to train MNIST for 1000 epochs using AVB with simple Neural net discriminator: 11.3 hrs
Time taken to train MNIST for 1000 epochs using AVB with RKHS discriminator: 14.7 hrs

### 8.3.4   Existing Assets

We followed the official implementation of Adversarial Variational Bayes [7] at
`https://github.com/LMescheder/AdversarialVariationalBayes`

## 10   Societal Impacts

We discuss possible negative impacts in two categories: 1) Impact of theoretical contribution, 2) Impact of applications

**Societal Impact of theoretical contribution:**   The main theoretical contribution of the paper is its connection between reliable/stable estimation and complexity analysis of the discriminator function space. In its general form, this contribution does not, by itself, pose any negative societal impact. Rather, it is about stabilizing algorithms. So, it contributes towards more robust and stable algorithms, and may help in developing more secure applications. We do not foresee any negative societal impacts in safety and security of human beings and automatic systems, human rights, human livelihood or economic security, environment. We do not see it causing theft, harassment, fraud, bias or discrimination.

**Societal Impact of possible applications:**   As demonstrated in the experiment section, this work can be applied to information theoretic applications that require mutual information or KL divergence estimation. For example, it has been used in generative modeling like variational autoencoder, variational Bayes or in stabilizing generative adversarial networks (GANs). These generative modeling techniques are, by themselves, quite general and can have numerous applications, including the ones with negative impacts. By helping in accurate estimation of KL divergence and by providing theoretical analysis, this work is contributing to develop stronger generative models and by extension could be indirectly helping in their negative uses. In that aspect, we appeal everyone using the algorithms and ideas in this paper to be thoughtful and responsible in their use.