# OpenReview forum: "Reliable Estimation of KL Divergence using a Discriminator in Reproducing Kernel Hilbert Space"
_NeurIPS.cc/2021/Conference — NeurIPS 2021 Spotlight_

### Official Review · Reviewer_mM61 · 2021-07-08

**Rating:** 8
**Confidence:** 3

**Summary:**

An improved method for estimating the KL divergence between two distributions based on samples. To estimate the log density ratio, the authors proposed to train a discriminator while constraining the family of the discriminator function. It constructs the function using the defined in F. Bach (2017), which is a convolution between two network functions. The authors did a careful (though no very thorough) performance analysis of this method, and demonstrated the performance in simple datasets for three different tasks involving the KL divergence.

**Limitations And Societal Impact:**

While the theory might be clear, the experiments were less than sufficient. They were mostly simple Gaussian distributions, and MNIST only appears to demonstrate stability.

**Main Review:**

The idea of constructing a constrained RHKS function for this density ratio estimation appears to be new. It provides a different formulation of the deep kernels used in supervised and unsupervised settings. The analysis method uses standard theoretical methods, such as concentration inequalities and covering numbers. Though some analyses involved much deeper derivations/proofs than others, and the experiments were not yet strong enough, I think the method and theoretical contributions are clear.

The written quality needs some improvement. For example, in eqn (5), $\psi$ is only defined later, and $\tau$ is never defined.

Here are some detailed comments:
I cannot fully follow the logic in the proof of Lemma 1, and there appear to be some (minor) inconsistencies. It also appears that what would be useful is the contrapositive of Eqn (17), i.e. $\ell_z(f_j)\le \epsilon \Longrightarrow \dots le 2\epsilon$, and then we can proceed by writing $\text{Prob}(\dots \le 2\epsilon)$ rather than a  $\ge$ statement. Also, is $\eta$ set to $\frac{\epsilon}{2\sqrt{S_K}}$ (in the proof) or $\frac{\epsilon}{4\sqrt{S_K}}$ (in the main text)?

Line 219. How's the norm of g related to R in Theorem 2?

There are a few questions regarding the experimental results.
1. Figure 1: top row shows a small bias for the blue clusters. Does this never go away? The bias in the bottom row also doesn't go away.
2. Figure 3: the neural network decoding seems bad at the end of training, but before the loss blows up, there seems to be a relatively stable region. How does the model at that stage perform in terms of decoding? Also the y axis are not equal, making the results a little hard to compare. Also, although the authors reminded the readers that the point is on stability, but is the proposed method reliability and effective, in terms of standard metrics like the log-likelihood or sample quality (FID, KID)?

**Time Spent Reviewing:**

4 hours

---

> ### Author Response · Authors · 2021-08-10
> **Proof of Lemma 1, KL curve and FID scores, value of $\eta$, norm of $g$,**
>
> Thank you for your detailed feedback. We appreciate that you went through our work in detail, including the proofs in the supplementary material. We are very thankful for your positive feedback and suggestions to improve experiments, writing and clarity of proofs.
>
> ### Q. ... cannot fully follow the logic in the proof of lemma 1
>
> We clarify the confusion in three steps: i) clarify eq.(17) ii) how eq.(18) is obtained from (17), iii) explain eq.(19).
>
> i) Clarification of eq.(17):
> Within a disk, $D_j$, we have , $||f-f_j|| \leq \eta$
> then from (16), we have $|\ell_z(f)- \ell_z(f_j)|\leq 2\sqrt{S_K}||f-f_j||_{\mathcal{H}_K} \leq 2\sqrt{S_K}\eta$. Now, we choose $\eta = \frac{\epsilon}{2\sqrt{S_K}}$.  Plugging in $\epsilon = 2\sqrt{S_K}\eta$ in the previous equation, we obtain, $|\ell_z(f)- \ell_z(f_j)|\leq \epsilon$.
> This is true for any $f$ within the disk $D_j$.
>
> Now, we show $\ell_z(f_j) \geq |\ell_z(f)| - \epsilon$. To show this, $|\ell_z(f)| = |\ell_z(f) - \ell_z(f_j) + \ell_z(f_j)| \leq |\ell_z(f) - \ell_z(f_j)| + |\ell_z(f_j)|$ using triangle inequality. Therefore, $|\ell_z(f)| \leq \epsilon + |\ell_z(f_j)|$, from which we obtain $\ell_z(f_j) \geq |\ell_z(f)| - \epsilon$. From this statement, it is easy to see that, for any $f \in D_j$, if $|\ell_z(f)| \geq 2\epsilon$, then it must be true that $|\ell_z(f_j)| \geq \epsilon$, which is precisely eq.(17).
>
> ii)Obtaining eq.(18) from eq.(17):
> Let us denote the left hand side and right hand side of eq.(17) as events A and B; i.e.,
> $\underset{f\in D_j}{sup}{| \ell_z(f)| \geq 2\epsilon}$ is denoted by $A$ and $|\ell_z(f_j)| \geq \epsilon$ as B.
> Then eq.(17) says $A$ implies $B$. That means whenever A happens, B also happens, but not vice versa. Therefore, represented as a set, set A is possibly smaller than B, and furthermore, A is a subset of B. When A is a subset of B, we have $Pr(A) \leq Pr(B)$ by using the Monotonicity property of probability measure.
> Using Hoeffding's inequality, we can further obtain, $Pr(B) = Pr(|\ell_z(f_j)| \geq \epsilon) \leq e^{\frac{-m\epsilon^2}{2M^2}}$. Since $Pr(A) \leq Pr(B)$, we obtain, $Pr(A) \leq e^{\frac{-m\epsilon^2}{2M^2}}$
> which is exactly eq.(18).
>
> iii) Explanation of last line after eq.(19):
> Eq.(19) is of the form $Pr(X\geq 2\epsilon) \leq a$. We take the probability of its complement. From eq.(19), therefore, we can obtain $Pr(X \leq 2\epsilon) \geq 1-a$, which would be $
> Pr(\underset{f\in \mathcal{H}}{sup}{| \ell_z(f)| \leq 2\epsilon} )\geq 1- 2\mathcal{N}(\mathcal{H},\frac{\epsilon}{2\sqrt{S_K}})e^{-\frac{m\epsilon^2}{2M^2}}
> $
> Then, we finally set $\epsilon = \epsilon/2$ to obtain the last line after eq.(19)
>
> ### Q. Is $\eta$ set to $\frac{\epsilon}{2\sqrt{S_K}}$ or $\frac{\epsilon}{4\sqrt{S_K}}$ ?
>
> In proving eq.(17), we choose $\eta = \frac{\epsilon}{2\sqrt{S_K}}$. But, later in proving the last line after eq.(19), we again set $\epsilon = \frac{\epsilon}{2}$ as we explained in the previous question. So, the combined effect is that we choose $\eta = \frac{\epsilon}{4\sqrt{S_K}}$ as stated in the main text.
>
> ### Q. How is the norm of $g$ related to $R$?
>
> The RKHS norm $||f||\_{\mathcal{H}\_K}$ is related to the norm of $g$, $||g||\_{\mathcal{L}\_2(d\tau)}$ by the following relation: $||f||^2\_{\mathcal{H}\_K} \leq ||g||^2_{\mathcal{L}\_2(d\tau)}$, where $d\tau$ is the probability measure on the domain $\mathcal{W}$ of $g$ as defined in Theorem 1. This is one of the important result due to Bach et al.[15]. $R$ is nothing but the radius of the RKHS ball under consideration, so $R = ||f||\_{H\_K}$. Hence, we use $||g||\_{\mathcal{L}\_2(d\tau)}$ as an upper bound on $R$ and minimize the upper bound during optimization. This relation between norm of $g$ and RKHS norm of $f$ is stated in Theorem 1, but it seems to be easily missed. So, we will explicitly explain this relation in the main text in our final version to enhance clarity.
>
> ### Q. KL curve before unstable epochs
>
> We used high range in the y-axis for the neural net discriminator (left figure) to highlight the fact that after around 600 epochs, the discriminator does get unstable. In the region before 600 epochs, the KL divergence with the neural net discriminator is to some extent stable although not as smooth as with the regularized RKHS discriminator. It still jitters a little bit.
>
> Some sample images during this time is shown in Fig 3(a): top row shows that decoding is not that terrible even for the unregularized neural network discriminator. To make this comparison more precise and quantitative, following your suggestion, we have computed FID score (smaller the better) between reconstruction and ground truth after each epoch.  Below are the FID scores.
>
>  Epoch  $ \hspace{0.7cm} 100   \hspace{0.5cm} 200  \hspace{0.5cm}   300  \hspace{0.5cm}  400  \hspace{0.5cm}   500   \hspace{0.5cm}  600   \hspace{0.5cm}  700   \hspace{0.5cm}  800 \hspace{0.5cm}    900 $
>
>  NNet  Disc. $49.64  \hspace{0.2cm}43.25 \hspace{0.2cm} 47.52  \hspace{0.2cm}44.29  \hspace{0.2cm}51.31 \hspace{0.2cm} 52.45  \hspace{0.2cm}148.9  \hspace{0.2cm}157.5  \hspace{0.2cm}261.9 $
>
>  RKHS Disc.  $37.58  \hspace{0.2cm}33.18 \hspace{0.2cm} 31.46 \hspace{0.2cm} 31.39  \hspace{0.2cm}30.36 \hspace{0.2cm} 29.37 \hspace{0.2cm} 28.17\hspace{0.2cm} 28.18 \hspace{0.2cm} 28.17$
>
> A couple of observations are in order. Even for the neural net architecture the FID score is below 50 upto 400 epochs and it is fluctuating. Thus, before the discriminator blows up, the reconstruction is okay. However, this score is worse than the RKHS discriminator which has scores in the range of 30-40 upto epoch 400. For the neural network discriminator, the score increases (worsens) in the range of 50 for epochs 500 and 600. Once the epoch reaches 700, the discriminator becomes completely unstable and FID score shoots up reaching 262 at epoch 900.
>
> On the other hand, for the RKHS discriminator, the score steadily and smoothly decreases as the epoch increases reaching the best 28.17 at epoch 900.
>
> This experiment demonstrates that the proposed RKHS discriminator with norm regularization is both reliable and effective in terms of standard metric like FID.
>
> ### Q.  Bias doesn't go away
>
> It is true that there seems to be a small bias in the experiment (Fig. 1(a,b)) even though our estimates are significantly better than using Neural network discriminator without regularization.
> We believe this might have to do with inherent estimation bias when we are dealing with finite amount of data. Even though we have shown that the estimate is consistent when the data goes to infinity; when data is finite, some bias may be inevitable.
>
> ### Q. $\psi$ and $\tau$ are not defined appropriately
>
> Thank you for your feedback. We will take this advice and appropriately define all the variables after Theorem 1.

---

> > ### Comment · Reviewer_mM61 · 2021-08-22
> > **Thanks for the explanation**
> >
> > The authors provided a very satisfactory response, and I appreciate their clarity in the explanations. The new experimental results more strongly support their methods. This is a good contribution to the field. So I will raise my score and hope to see this paper published. I also encourage the authors to revise the writing of the paper with the clarity demonstrated in their responses.

---

> > > ### Author Response · Authors · 2021-08-24
> > > **We will include revision**
> > >
> > > Thank you. We will include the revision in the final version.

---

### Official Review · Reviewer_2N7B · 2021-07-13

**Rating:** 7
**Confidence:** 3

**Summary:**

The authors consider the problem of KL divergence estimation. Modern machine learning methods for KL divergence estimation make use of neural networks as function approximators. The authors assert that many of the challenges with using such methods is the uncontrolled complexity of the function space. They propose to resolve this by constructing their neural network functions to be RKHS functions, and control the RKHS norm. They prove learning-theoretic results about their method, showing that it is consistent and exhibits controllable fluctuations around its mean (deviation-from-mean error).

**Limitations And Societal Impact:**

Yes

**Main Review:**


The paper is nicely written and introduces the problem well. I haven't read the proofs in detail but I believe the authors make a solid technical contribution. They are honest about the limitations of their method (discussion of assumptions and paragraph at end) and do not overstate their results, which is refreshing. They provide sketches of proofs in the main text to provide intuition / high level understanding of the results.

I can't comment on the novelty of the idea; I am unaware of work that tries to use RKHS norms to control neural network function complexity in this way, but I am not an expert in this field and am not up to date with the latest literature. If this is a novel idea then I find it quite exciting and is worthy of merit in its own right (regardless of the rest of the paper). If it is not novel, appropriate citations should be added.

I'm generally happy with it but have a few points I'd like to be addressed. I've given "weak accept" but would be willing to raise this rating if the authors can address them:

- the results are all cast in terms of the L2(\tau) norm of g. How does this relate to the RKHS norm of f? At the beginning of the paper, I thought you would control the RKHS norm of f -- I think this is implicitly being done, but would be great if you could make this more explicit / comment on this.
- variance analysis: I understand that theorem 3 only applies under the strong assumption of the deviation-from-mean being Gaussian. Is it not possible to make some statements about how the variance decays under some other weaker assumptions? If not, are there (perhaps pathological) cases where the variance does not decay?
- at the beginning you claim that the existing methods suffer due to having uncontrolled function complexity. Do you have any evidence to support this claim? Do these other methods not use regularisation in practice?


Non-technical issues:
- Figures: I think you're converting your figures to PNG or JPG before embedding them into the document. Much better is to save them as PDFs -- that way they are not blurry when you zoom in on a computer.
- I think you are maybe just writing KL in math mode, but if you use use add this latex command: \newcommand{\KL}{\mathrm{KL}} and then use \KL it may look better depending on your taste (feel free to ignore this if you don't like it though).



Other detailed comments:
- line 54: gap before (RKHS)
- line 138: "w ~ N(O, \gamma I)" -> is this saying that \tau is the normal distribution? Could you state that explicitly if so?
- line 140: could you maybe make it explicit here that \phi (and I guess also g?) will be implemented as neural networks?
- line 144: "both the RKHS kernel and its norm" -> "both the RKHS kernel and the norm of f"
- You don't really discuss the RKHS norm of f directly. Could you at least briefly mention it here, in particular how it relates to \phi and g?
- line 151: both \tau and h have already been used, but I don't think the occurrences in this line are referring to either of them?
- line 166: what does M-bounded mean?
- line 194: approximately what kind of order of magnitude do you expect the positive part inside the exp to be? (The practical usefulness of the results does depend on this.)
- equation (14): I guess the \sigma on the left hand side is supposed to be the standard deviation of  the estimator? I
- line 264: typo "Guaussian"


[Edit: raised score by 1 point after authors addressed comments in review]

**Time Spent Reviewing:**

4

---

> ### Author Response · Authors · 2021-08-10
> **Relation between $||g||$ and $||f||$, Novelty, Variance analysis, Complexity control**
>
> We would like to thank you for your detailed feedback on our work, technical, non technical and related to writing and presentation. We are also thankful for the encouraging comments on the novelty and merit of our work. We feel encouraged and excited.
>
> ### Q. Novelty of the idea
>
> To the best of our knowledge, there is no previous work that tries to model a neural network function explicitly such that it lies on an RKHS space. Also, we have not seen any work that try to look at discriminators from the perspective of complexity of their function space and its role in stability. Therefore, we are also excited that this could open multiple interesting research directions for researchers in the field. We are very thankful for your encouragement.
>
> ### Q. Relation between L2 norm of $g$ and RKHS norm of $f$
>
> The RKHS norm $||f||\_{\mathcal{H}\_K}$ is related to the norm of $g$ , $||g||\_{\mathcal{L}\_2(d\tau)}$ by the following relation: $||f||^2\_{\mathcal{H}\_K} \leq ||g||^2\_{\mathcal{L}\_2(d\tau)}$, where $d\tau$ is the probability measure on the domain $\mathcal{W}$ of $g$ as defined in Theorem 1. This is one of the important results due to Bach et al.[15].  Hence, we use $||g||\_{\mathcal{L}\_2(d\tau)}$ as an upper bound on $||f||\_{\mathcal{H}\_K}$ and minimize the upper bound during optimization. This relation between norm of $g$ and RKHS norm of $f$ is stated in Theorem 1, but it seems to be easily missed. So, we will explicitly explain this relation in the main text in our final version to improve clarity.
>
> ### Q. Variance analysis: ...is it not possible to make statements about how the variance decays under some other weaker assumptions?
>
> This is an important question for discussion. So far, we know that we can do similar analysis if the estimate is distributed as an exponential distribution. The key idea is how the cumulative distribution function (CDF) is related to the variance. To clarify this point further, let's look at the eq.(33) in the proof of Theorem 3 in supplementary material:
> $1 - \Phi_{\mu, \sigma}(\mu + \epsilon) \leq 2\exp\Bigg[\left( \frac{4RC_s\sqrt{S_K||\mathscr{L}_s||}}{\epsilon} \right)^{\frac{2n}{h}}-\frac{m\epsilon^2}{4M^2}\Bigg]$
>
> This equation connects the CDF to the variables like $S_K$, $R$, $\mathscr{L}\_s$ of the discriminator function space and does not yet assumes anything about the shape of the distribution. For a Gaussian distribution, we plug in CDF of a Gaussian distribution, $\Phi_{\mu, \sigma}(\hat{x}) = \frac{1}{2}\Big[1 + \text{erf} \big(\frac{\hat{x}-\mu}{\sigma \sqrt{2}}\big)\Big]$. For any other distribution, we can simply plug in the expression for the CDF, and carry out analysis similar to the proof of Theorem 3. However, a key factor that determines the behavior between the variance and the discriminator complexity is how variance appears in the CDF expression.
> For example, in both Gaussian distribution and in exponential distribution, we know that the relation between CDF function and variance is inversely related - note that the standard deviation, $\sigma$ appears in the denominator of Gaussian CDF.
> We believe that as long as this inverse type of relation between CDF and variance holds, we can obtain statements like Theorem 3 for other distributions as well. Consequently, in such distributions, the behavior that the variance decays as we penalize the complexity should hold. However, the behavior of the CDF function and its relation to variance depends on the exact shape of the probability density function (PDF) or CDF function of the estimate. Without the knowledge of the shape of the CDF or PDF, it is not possible to make statements about the decay of the variance. By changing the relation between CDF and variance, one may even be able to construct a pathological case where the variance doesn't decrease, but that would be too artificial.
>
> ### Q. Uncontrolled complexity in other methods
>
> In the original papers like AVB and VDM, they neither consider the  discriminator from the function space perspective nor try to control its complexity. However, to help in several practical aspects of training they apply other techniques. For example, gradient clipping has been suggested in VDM. Similarly, in AVB, they modify the KL divergence loss by using an adaptive Gaussian distribution, a method termed as 'adaptive contrast'. Additionally, they also apply other design choices in the discriminator, like adding a direct skip connection in the discriminator. While these methods may help in training, it is not clear if they actually control the complexity of the discriminator. When it comes to explicit complexity regularization, we are not aware of any method that explicitly regularize the complexity of the discriminator. There is also another problem in trying to control the complexity of a neural network discriminator: it is difficult to come up with an appropriate/meaningful notion of complexity. One of the reasons we constructed $f$ in RKHS is that it enables us to use well-defined complexity measure of the RKHS function space.
>
> ### Q. Nontechnical issues
>
> The images were directly plotted using matplotlib in python. We will save images in pdf format as you suggested to increase image quality. Also, we will increase the quality of those plots by setting dpi high while saving.
>
> Using '\mathrm{ KL } ' instead of 'KL' in the math mode seems like a good idea. We will follow your suggestion.
>
> ### Q. Other detailed comments
>
> line 138: Yes $\tau$ is a Gaussian distribution and we will explicitly mention it in our final version.
>
> line140: We will make it explicit that $\phi$ and $g$ are neural networks.
>
> line We will explicitly mention the relation between RKHS norm and $g$ in the paragraph after Theorem 1.
>
> line 166: M-bounded means the infinity norm of the function $f$ is upper bounded by a number $M$
>
> line 194: Unfortunately it is difficult to estimate numerical value of this quantity as it involves some constants like $C_p$ which are not really known and it is very difficult to compute norms like $||\mathscr{L}_p||$.
>
> equation 14: Yes, $\sigma$ is the standard deviation of the estimate. We will correct this in our final version.
>
> We will make appropriate changes in line 54, 144, 151 and 264 as you have suggested in the final version of our paper.

---

> > ### Comment · Reviewer_2N7B · 2021-08-24
> > **Response**
> >
> > Many thanks for the detailed response to the review! The reply is very clear and detailed. I'm happy that my concerns have been addressed so will raise my score.
> >
> > (I appreciate that it can be difficult with space, but I'd be happy if the authors can include a condensed version of the discussion about the variance analysis above in the updated paper, or at a minimum in the appendix. Thanks!)

---

> > > ### Author Response · Authors · 2021-08-24
> > > **We will include discussion**
> > >
> > > Thank you. We will definitely update the discussion in the supplementary material, and to the extent possible, will try to include a concise paragraph in the main paper.

---

### Official Review · Reviewer_kqvB · 2021-07-14

**Rating:** 7
**Confidence:** 3

**Summary:**

This papers studies the problem of estimating KL divergence from samples via variational methods with neural network discriminators. They propose a novel way to construct the discriminator in some RKHS and prove the consistency of the resulting estimator. By penalizing the RKHS norm of the discriminator, they reduce the variance of their estimator and stabilize the training.

**Limitations And Societal Impact:**

See comments above for suggestions.

**Main Review:**

**Originality**

Although their idea of constructing a function in the RKHS is not new, the usage of this construction in GAN-type objective for KL estimation seems to be novel. Related work is adequately cited and it is clear how their method differs from previous ones.

**Quality**

The submission is technically sound and most of the claims are well supported. Theoretical results are solid and informative. Experimental results are reasonable and show the improvements of the proposed method over previous ones. Below are several comments/questions I have:

1. In Abstract and line 40, they claim that methods with a neural-net based discriminator suffers from unreliability or instability. However, they do not include the two methods, VDM and MINE, cited in the paper in the experiments. I would like to see results that can support the claim.
2. In Theorem 2, the quantities $C_p$, $S_p$, $\mathscr{L}_p$ are not discussed. How are they related to the architecture of $\phi_\theta$ and $\gamma$? Are they always finite?
3. The meaning of $\text{sup}_{K_\theta}$ is unclear (e.g., in Theorem 2). Does that include all possible architectures of $\phi_\theta$ and all values of $\gamma$? If you maximize over $\gamma$, then the finiteness in the above comment may be an issue.
4. Since what you are advertising is the regularized version of the estimator, it would be better to include theoretical analysis for the consistency of this regularized estimator.
5. Spectral normalization is used in the proposed discriminator but not in the baseline. This seems to be unfair. Do you have an explanation about it?

**Clarity**

The paper is clearly written and easy to follow.

**Significance**

The results are likely interesting and useful to the community. The idea of constructing a function in RKHS and controlling its complexity could be used in GANs in general.

**Time Spent Reviewing:**

6

---

> ### Author Response · Authors · 2021-08-10
> **Experiment on other KL estimation methods, Finiteness of $C_p$, $S_p$, $\mathscr{L}_p$, Consistency of estimator, Spectral normalization**
>
> We would like to thank you for being generous with your time, going through the details of our work and providing us insightful feedback. We are also encouraged to know that you see the potential of our work to have a broader and more general impact beyond KL divergence estimation, for example, in GANs.
>
> ### Q. $C_p, S_p, {L}_p$. How are they related to $\phi$ and $\gamma$
> $C_p$ is a constant not depending on $S_p$, $L_p$ or $R$,  and it is always finite. See https://home.ttic.edu/~smale/papers/math_foundation_of_learning.pdf page 20, example 4 and proof of Theorem D in pages 50-51 for more details. They use results from [Edmunds and Triebel 1996], who prove that there exists some constant $C$, to obtain entropy number in Sobolev spaces.
>
>  $S_p$ and $L_p$ are related to $\phi$, $\gamma$ through the kernel $K_{\theta_p}$. Note that $K_{\theta_p}$ is the optimal kernel corresponding to the optimal neural network parameter $\theta_p$. $S_p$ is defined as
> $S_p=\underset{x,t}{sup} \hspace{0.1cm} {K(x,t)} = \underset{x,t}{sup} \hspace{0.1cm} \gamma \phi_{\theta_p}^T(x)\phi_{\theta_p}(t)$.
> Therefore, it is directly related to the network $\phi_{\theta_p}$. We show that this is finite due to Lipschitz constraint on $\phi_{\theta_p}$ as described in Assumption A2. The proof is provided in the Supplementary material page 2.
>
> $||\mathscr{L}\_p||$ is upper bounded as follows:
> $||\mathscr{L}\_p|| \leq \rho(\mathcal{X})\sum\_{|\alpha|\leq h/2}\underset{x,t \in \mathcal{X}}{sup}(D^{\alpha}_x K\_{\theta_p}(x,t))^2$
>
> $= \rho(\mathcal{X})\sum\_{|\alpha|\leq h/2}\underset{x,t \in \mathcal{X}}{sup}(D^{\alpha}\_x \gamma \phi\_{\theta_p}^T(x)\phi\_{\theta_p}(t))^2$,
> where $D^{\alpha}\_x$ is an $\alpha$ order differentiation operator.
> Therefore, the finiteness of $||\mathscr{L}\_p||$ depends on the finiteness of the higher order derivatives of $\phi_{\theta_p}$. $||\mathscr{L}\_p||$ is finite if the higher order derivatives exist and are finite. It means if the function $\phi_{\theta_p}$ is differentiable, and gradients are not infinite, then $||\mathscr{L}\_p||$ is also finite and well-behaved. For example, when the activation function inside the neural network is differentiable, the derivative of $\phi_{\theta_p}$ is finite and $||\mathscr{L}\_p||$ is also finite. In the case of ReLU activation function, however, this is more subtle because, strictly speaking, ReLU is not differentiable at the origin. There may, however, be a way to use subgradients or define gradients in a different way consistent with the backpropagation (after all, backpropagation through ReLU is commonly available, so in some sense the gradient of ReLU is still well-defined). We have noted this difficulty in the limitation section of our paper and left the more rigorous work in this direction for the future.
>
> ### Q. Is sup $K_{\theta}$ over $\phi$ and $\gamma$ both?
> $\underset{K_{\theta}}{sup}$ is only with respect to architectures $\phi_{\theta}$. We treat $\gamma$ as a hyperparameter and is treated as a constant in the optimization.
>
> ### Q. Results showing VDM and MINE are unstable
> We have carried out these experiments and reproduced the results as per your suggestion.
>
> Experiment details:
> The experimental setup and discriminator architecture are the same as with other experiments in main paper (like Fig. 1). We randomly sample (2500) points from each of the two distributions and run experiments on these fixed samples to simulate a scenario, where data do not come infinitely from the source. With the same simple architecture and setup as our other experiments, we ran VDM and MINE.
>
> For the case where the ground truth value is 13.8, VDM gives unreasonably high estimates as follows (on 10 independent runs):
>
> $836.43, \hspace{0.3cm} 459.26,\hspace{0.3cm} 589.36, \hspace{0.3cm}1158.93, \hspace{0.3cm}386.1, \hspace{0.3cm}3232.4, \hspace{0.3cm}1067.3, \hspace{0.3cm}263.2, \hspace{0.3cm}1010.7$
>
> For ground truth value of 38, the estimates go high in the order of $10,000$.
>
> Similarly, we ran experiments for MINE. In this experiment, the value of KL divergence steadily increases and after some time, it starts to give a 'Not a number' token, which we consider as being unstable.
>
> In the beginning of our project, we did these experiments and realized that in this setting, these methods are unstable. We also verified that other works in the literature have reported such cases of unreliable estimates [8,9]. Therefore, we decided to shift our focus only on AVB type of objective, which was slightly better behaved than VDM and MINE at the moment. We tried to understand how the discriminator complexity plays a role in AVB type objective, and, to that end, developed a theory that could shed light on the stability/reliability behavior of those objectives (eq. 3,4 in the main paper). Since the objective functions of VDM/MINE are different from the objective we analyzed, our theoretical analysis is not directly relevant to VDM/MINE. Second, the results were unstable. Therefore, we did not include the numerical results of VDM/MINE in our main paper and simply commented that they were unstable. We also did not carry out extensive experiments after our initial experiments in VDM/MINE. However, we believe that not controlling the complexity of the discriminator could be a common cause of instability in discriminator based methods and could be directly related to the unstable behavior of VDM/MINE approaches too.
>
> ### Q. Consistency of the regularized estimator?
> Theorem 4 actually talks about the consistency of the regularized estimator. In the statement of Theorem 4, eq.(13) is used as the optimal function, $f^m_h$, which optimizes the regularized objective. And then we show that $|KL_m(f^m_h)-KL(f^*)| \to 0$, i.e. the KL estimate obtained from the regularized objective approaches the true KL estimate. Theorem 4 in the supplementary material has more details and proof of consistency.
>
> ### Q. Spectral normalization is not used in the baseline
> Spectral normalization is one of the important assumptions based on which we develop our theory. Specifically, using spectral normalization, we show that the quantities like $S_p$ and $||g||^2$ are finite (see Supplementary materials page 2). Intuitively, we believe spectral normalization helps to ensure the smoothness of the discriminator and perhaps helps in controlling complexity and in stabilization. On the other hand, the baseline methods do not take into account the complexity of the discriminator or consider its smoothness. Therefore, we simply followed the proposed algorithms of baseline methods and did not use spectral normalization with them.
>
>
> ### Reference
>  Edmunds, D. and Triebel, H. Function Spaces, Entropy Numbers and Differential Operators. Cambridge University Press, 1996.

---

> > ### Comment · Reviewer_kqvB · 2021-08-25
> > **Thank you for your response**
> >
> > Thank you for your detailed response. My questions were answered so will I raise my score. I encourage the authors to make changes to the paper based on the response here.

---

### Decision · Program_Chairs · 2021-09-27

**Decision:**

Accept (Spotlight)

**Comment:**

Converging reviews regarding the acceptance of the paper after a fruitful discussion between reviewers and authors, where the latter took time to clarify all the questions raised.
The recommendation for this paper is to accept it, counting on the authors to take into account the outcomes of the discussions with all the reviewers in the final version of the work (preferably in the main text rather than in the supplementary material).